# Expanding the terpene biosynthetic code with non-canonical 16 carbon atom building blocks

Codruta Ignea [1,5], Morten H. Raadam [1], Aikaterini Koutsaviti[2], Yong Zhao [1], Yao-Tao Duan[1], Maria Harizani[2], Karel Miettinen [1], Panagiota Georgantea[2], Mads Rosenfeldt[1], Sara E. Viejo-Ledesma[1], Mikael A. Petersen[3], Wender L. P. Bredie [3], Dan Staerk [4], Vassilios Roussis[2], Efstathia Ioannou [2] ✉ & Sotirios C. Kampranis [1] ✉

Humankind relies on specialized metabolites for medicines, flavors, fragrances, and numerous other valuable biomaterials. However, the chemical space occupied by specialized metabolites, and, thus, their application potential, is limited because their biosynthesis is based on only a handful of building blocks. Engineering organisms to synthesize alternative building blocks will bypass this limitation and enable the sustainable production of molecules with non-canonical chemical structures, expanding the possible applications. Herein, we focus on isoprenoids and combine synthetic biology with protein engineering to construct yeast cells that synthesize 10 non-canonical isoprenoid building blocks with 16 carbon atoms. We identify suitable terpene synthases to convert these building blocks into $C_{16}$ scaffolds and a cytochrome P450 to decorate the terpene scaffolds and produce different oxygenated compounds. Thus, we reconstruct the modular structure of terpene biosynthesis on 16-carbon backbones, synthesizing 28 different non-canonical terpenes, some of which have interesting odorant properties.

Isoprenoids (or terpenoids) are a large group of chemical compounds that participate in both basic and specialized metabolism and have diverse functions, ranging from membrane components (sterols) to hormones (steroids, abscisic acid, gibberellins, pheromones), signaling molecules (mono- and diterpenes), pigments (carotenoids), or defense compounds (diterpenes, saponins)[1]. Humankind relies heavily on isoprenoids for medicines (taxol, artemisinin), flavors (menthol), fragrances (geraniol, santalol, sclareol), colorants (carotenoids), and biopolymers (caoutchouc, gutta percha)[1].

Isoprenoid biosynthesis is a modular process (Fig. 1). Initially, two $C_5$ isomers, isopentenyl diphosphate (IPP) and dimethylallyl diphosphate (DMAPP), combine to form prenyl diphosphate precursors of different sizes by successive addition of five-carbon units. Each of these building blocks lays the foundation for an entire, distinct class of compounds, i.e. geranyl diphosphate (GPP) for monoterpenes ($C_{10}$), farnesyl diphosphate (FPP) for sesquiterpenes ($C_{15}$), triterpenes ($C_{30}$), and sterols, geranylgeranyl diphosphate (GGPP) for diterpenes ($C_{20}$) and carotenoids ($C_{40}$). Each class-specific building block gives rise to

[1]Biochemical Engineering Group, Plant Biochemistry Section, Department of Plant and Environmental Sciences, University of Copenhagen, Thorvaldsensvej 40, 1871 Frederiksberg C, Denmark. [2]Section of Pharmacognosy and Chemistry of Natural Products, Department of Pharmacy, School of Health Sciences, National and Kapodistrian University of Athens, Panepistimiopolis Zografou, Athens 15771, Greece. [3]Department of Food Science, University of Copenhagen, Rolighedsvej 26, 1958 Frederiksberg C, Denmark. [4]Department of Drug Design and Pharmacology, Faculty of Health and Medical Sciences, University of Copenhagen, Universitetsparken 2, 2100 Copenhagen, Denmark. [5]Present address: Department of Bioengineering, McGill University, McConnell Engineering Building, 3480 University, Room 350, Montreal, QC H3A 0E9, Canada. ✉e-mail: eioannou@pharm.uoa.gr; soka@plen.ku.dk

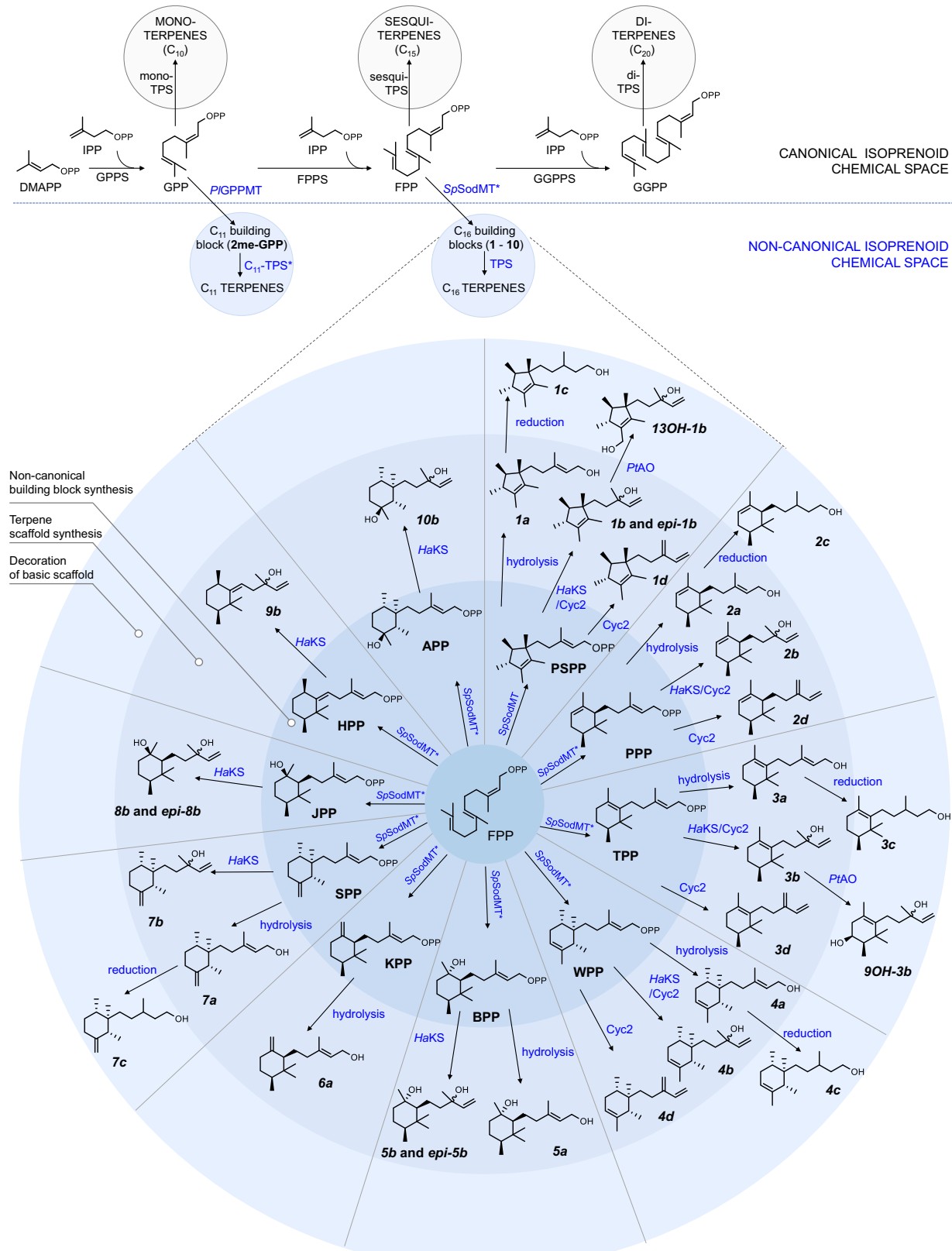

numerous hydrocarbon skeletons by the activity of terpene synthases[1]. Subsequently, decorating enzymes (cytochrome P450s, dehydrogenases, *O*-acetyltransferases, *O*-methyltransferases, etc.) expand the complexity of the terpene scaffolds to produce a broad diversity of compounds within each specific class (Fig. 1).

As the biosynthesis of isoprenoids is based on the $C_5$ isoprene unit, the different isoprenoid classes differ in their size by five carbon atoms. This creates a stratification in the possible chemical structures and restricts the complexity of isoprenoids to skeletons of 10, 15, 20, or higher multiples of five carbon atoms[1]. Only very few molecules with a

**Fig. 1 | Expanding the terpenoid chemical space with $C_{16}$ terpenes.** Canonical terpene biosynthesis is shown at the top (gray circles). IPP and DMAPP are condensed by prenyltransferases (geranyl diphosphate synthase (GPPS), farnesyl diphosphate synthase (FPPS), geranylgeranyl diphosphate synthase (GGPPS)) to form building blocks of 10 (GPP), 15 (FPP), or 20 (GGPP) carbon atoms. Subsequently, these canonical building blocks are cyclized by terpene synthases (TPSs) to form hydrocarbons with the corresponding number of carbon atoms. Previously, we expanded the chemical diversity of terpenoids by synthesizing $C_{11}$ molecules by engineering yeast cells to produce 2-methyl-GPP (2me-GPP)[17] (top, left blue circle). In this study, we further expand this diversity by engineering yeast to synthesize 10 non-canonical $C_{16}$ building blocks (**1-10**) through methylation of FPP by a dedicated methyltransferase (*Sp*SodMT) and its engineered variants (commonly denoting mutants engineered in this study as *Sp*SodMT*). To obtain non-canonical terpenes,

these building blocks were converted into $C_{16}$ tertiary alcohols (**1b–5b**, **7b–10b**) and $C_{16}$ olefins (**1d–4d**) by surrogate terpene synthases (KS and Cyc2). Moreover, $C_{16}$ primary alcohols (**1a–7a**) were produced by non-specific hydrolysis of $C_{16}$ diphosphates by yeast phosphatases, while their reduced counterparts (**1c–4c** and **7c**) were obtained by the action of yeast oxidoreductases. Further decoration of the $C_{16}$ core structures was achieved by a multifunctional cytochrome P450 (*Pt*AO), which used **1b** and **3b** as substrates to produce oxygenated $C_{16}$ compounds **13OH-1b** and **9OH-3b**, respectively. Thus, we engineer the biosynthesis of an entire class of non-canonical terpenes with increasing complexity by recapitulating the modularity of terpene biosynthesis using the non-canonical building blocks produced. The asterisk (*) denotes engineered enzymes. The chemical structures of the isolated compounds depict only their relative configuration, as determined on the basis of their NMR data (Supplementary Fig. 1, Supplementary Note 1).

carbon number that deviates from this rule have been identified and, in these rare exceptions, the biosynthetic mechanisms involved are not conserved[2–7]. Thus, the chemical space that lies between the $C_5$-based layers remains largely inaccessible to biological synthesis. Expanding the diversity of isoprenoids by engineering industrial microorganisms, like the yeast *Saccharomyces cerevisiae*, to produce isoprenoid building blocks with non-canonical sizes would dramatically increase the diversity of these highly valuable molecules and expand their potential industrial applications.

Systematic expansion of the chemical space occupied by biologically-synthesized isoprenoids requires to, first, establish the synthesis of non-canonical isoprenoid building blocks and, subsequently, reconstruct the downstream steps of isoprenoid biosynthesis by identifying or engineering enzymes (terpene synthases, cytochrome P450s, etc.) able to convert the alternative precursors into potentially valuable compounds. Several inspiring chemoenzymatic studies have shown that the substrate promiscuity of terpene synthases can be successfully exploited to synthesize non-canonical terpenoids using alternative, chemically synthesized, substrates[8–13]. Thus, it would be possible to identify or engineer terpene synthases to convert non-canonical substrates. In addition, terpene-oxidizing P450s also display considerable substrate promiscuity[14–16] and could be exploited to decorate the non-canonical terpene scaffolds produced by such an approach. However, engineering an organism to synthesize an additional class of terpenoids exclusively from sugar also requires establishing biosynthetic pathways for non-canonical prenyl diphosphate substrates that are dedicated to the specific compound group. As the first step in this direction, we achieved the systematic synthesis of $C_{11}$ terpenes in yeast[17]. We engineered yeast to produce a precursor with 11 carbon atoms, 2-methyl-GPP, by hijacking a methyltransferase from the cyanobacterial 2-methylisoborneol biosynthesis pathway, which specifically adds a methyl group to GPP, and engineered several monoterpene synthases to preferentially accept 2-methyl-GPP as substrate instead of GPP. As a result, we were able to synthesize several $C_{11}$ terpene carbon skeletons, establishing proof of concept that it is possible to expand the terpene chemical space by synthetic biology.

Extending this concept to synthesize non-canonical terpenoids with larger size will have considerable impact because structural complexity increases dramatically as the number of carbon atoms increases. Moreover, larger terpenoids, like sesqui- or di-terpenoids, have numerous applications and are frequently used as pharmaceutical agents, fragrances, or high-value chemicals. To identify potentially useful biosynthetic activities that would enable a specific and systematic approach for the biosynthesis of larger terpenoids, we searched for evidence of synthesis of non-canonical terpenes in different organisms. We identified a study reporting the presence of a terpene, named sodorifen, with an unprecedented structure of 16 carbon atoms, dominating the volatile product profile of the rhizobacterium *Serratia plymuthica*[18]. The biosynthesis of sodorifen was subsequently reported to involve a methyltransferase (*Sp*SodMT) that catalyzes the sequential methylation and cyclization of FPP to a

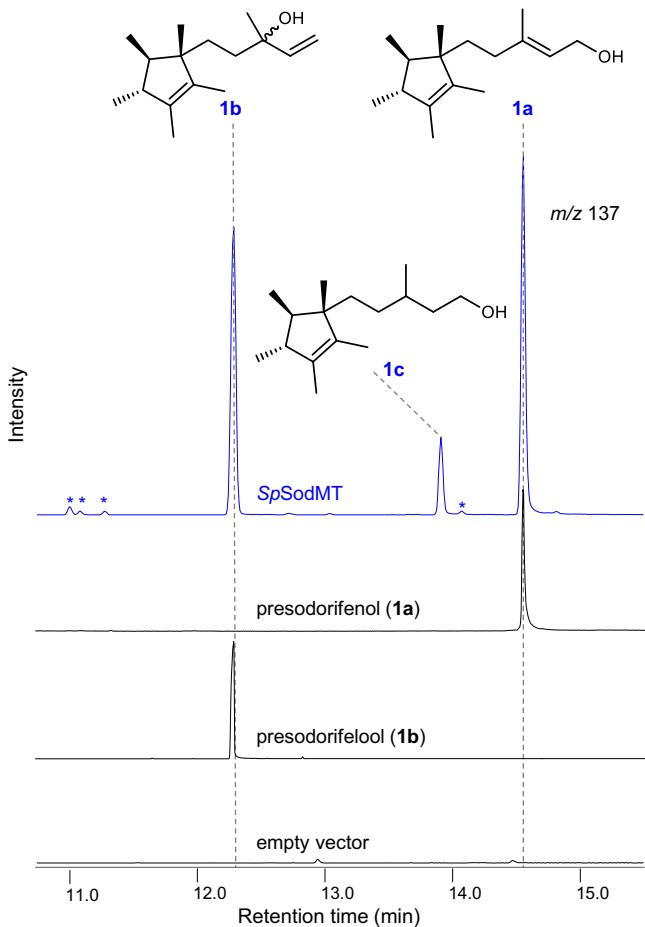

**Fig. 2 | Production of presodorifenyl diphosphate in yeast.** Expression of *Sp*SodMT in yeast resulted in the production of three alcohols, **1a**, **1b** and **1c** (blue), which were isolated and identified as presodorifenol, presodorifelool, and 2,3-dihydro-presodorifenol, respectively. Chromatograms in black show purified and structurally-characterized **1a** and **1b** compounds. None of these compounds was produced by yeast cells carrying the empty vector (also in black). Traces of four other compounds (*) were observed.

pentacyclic ring-containing 16-carbon atoms precursor, presodorifenyl diphosphate (PSPP), which is subsequently converted to sodorifen by a terpene synthase (*Sp*SodTPS)[19,20]. We decided to explore whether *Sp*SodMT could be hijacked for the synthesis of non-canonical isoprenoid building blocks in an engineered microorganism.

In this study, we introduce *Sp*SodMT in yeast and establish robust and orthogonal production of PSPP. We subsequently identify *Sp*SodMT residues responsible for product selectivity and engineer enzyme variants that synthesize nine additional $C_{16}$ prenyl

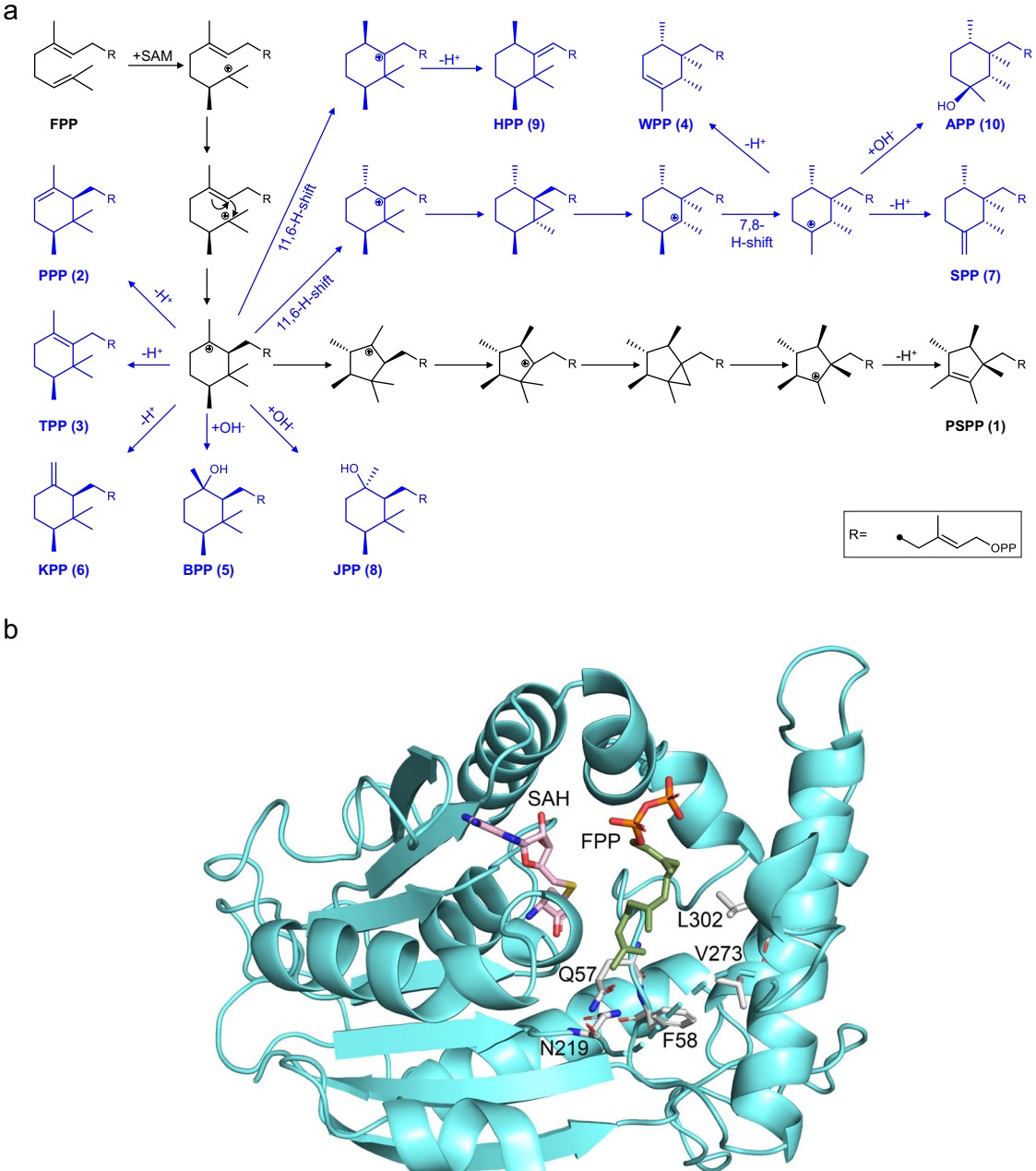

**Fig. 3 | _Sp_SodMT model and proposed mechanism. a** Proposed mechanism of the methylation-mediated cyclization reaction leading to the synthesis of the different C_{16} building blocks by _Sp_SodMT and its variants. The mechanism of PSPP synthesis proposed by von Reuss and co-workers[19] is shown in black. Proposed carbocationic cascades leading to the synthesis of PPP (**2**), TPP (**3**), WPP (**4**), BPP (**5**), KPP (**6**), SPP (**7**), JPP (**8**), HPP (**9**), or APP (**10**) are shown in blue. **b** Model of _Sp_SodMT with FPP and S-adenosyl-L-homocysteine (SAH). Key residues lining the cavity where cyclization of methylated FPP occurs and found to have an impact on product outcome are shown. Graphic constructed with PyMOL (Schrödinger, LLC).

diphosphate building blocks. By identifying terpene synthases that accept the non-canonical substrates, we utilize this collection of building blocks to produce different terpene scaffolds. Through the identification of one cytochrome P450 enzyme able to decorate the C_{16} scaffolds, we synthesize several oxygenated non-canonical terpenes. In total, we produce, isolate and structurally characterize 28 isoprenoid compounds with C_{16} carbon skeletons (Fig. 1). Moreover, we identify several of these compounds to be odor-active.

## Results
### Establishing methylation of FPP in yeast
To establish methylation of FPP in yeast, we introduced a yeast codon-optimized version of the methyltransferase _Sp_SodMT into a strain

engineered to produce high-levels of FPP (AM109[21]; Supplementary Data 1). Following GC-MS analysis of hexane extracts of _Sp_SodMT-expressing AM109 cells, two major products, **1a** and **1b**, and one minor one, **1c**, could clearly be detected (Fig. 2). We predicted **1a** and **1b** to be the primary and tertiary alcohols derived from the C_{16} product of _Sp_SodMT, presodorifenyl diphosphate (PSPP; **1**)[19], upon dephosphorylation. This was based on previous studies showing that yeast cells producing excess of prenyl diphosphates convert these compounds to the corresponding primary alcohol by the action of intracellular yeast phosphatases, such as Lpp1p and Dpp1p[22–24], or to the corresponding tertiary alcohol by hydrolysis in the acidic conditions of the yeast culture medium[25,26]. To confirm this prediction, we proceeded to isolate compounds **1a**, **1b**, and **1c** and elucidate their

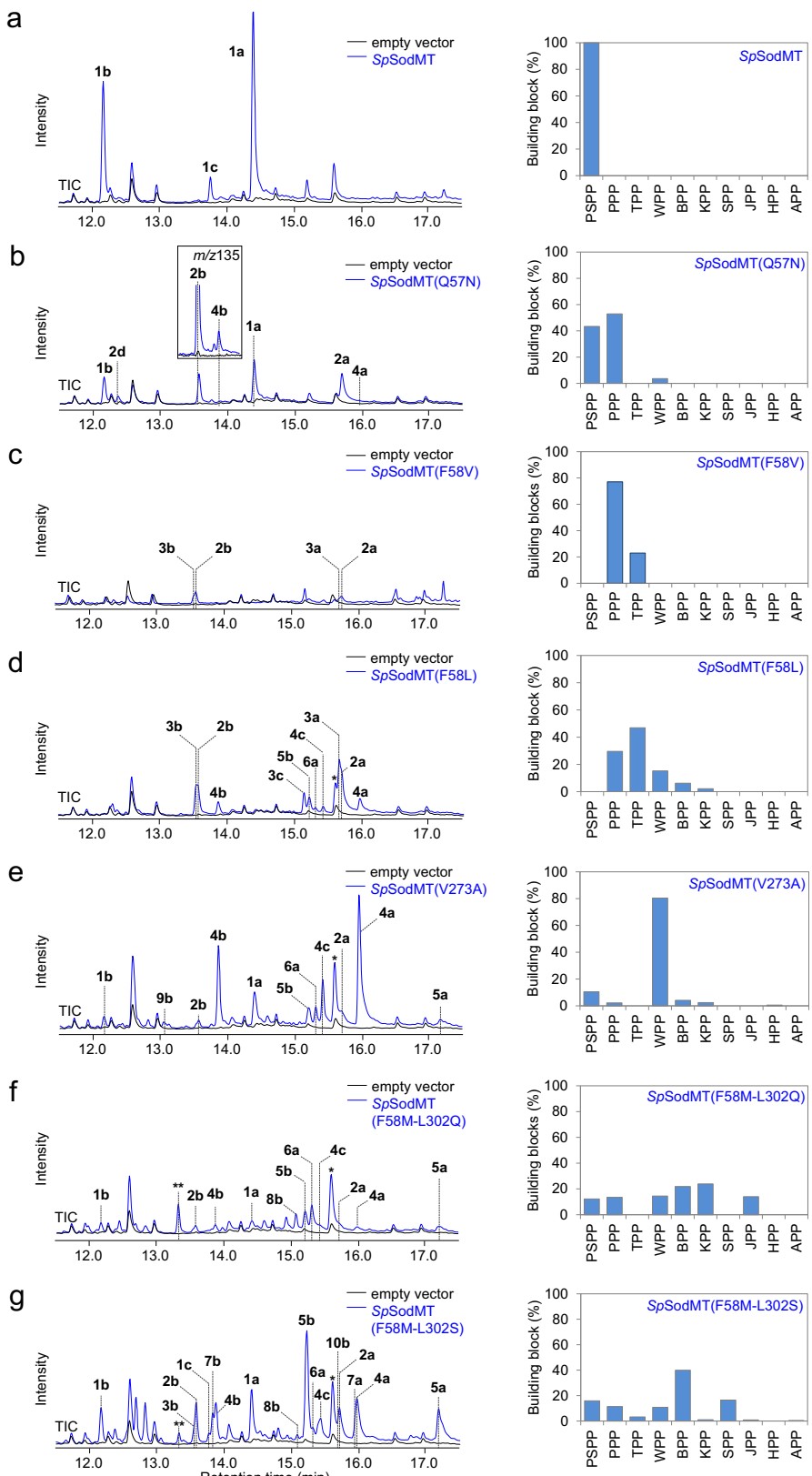

structures by analysis of their NMR spectroscopic data. Indeed, compound **1a** was identified as presodorifenol (or presodorifen[19]), while compound **1b** was found to be its tertiary alcohol counterpart, which we name here presodorifelool (Fig. 2 and Supplementary Table 1, Supplementary Fig. 1, Supplementary Note 1, Supplementary Data 2). Consistent with the proposed mechanism for the formation of the tertiary alcohol by hydrolysis, we also isolated and elucidated the structure of the epimeric form of **1b**, *epi*−**1b** (Supplementary Table 1, Supplementary Fig. 1, Supplementary Note 1, Supplementary Data 2). Compound **1c** was identified as the 2,3-reduced form of **1a**, in agreement with previous studies showing that, in *S. cerevisiae*, geraniol is converted to its 2,3-hydrogenated form, citronellol, by endogenous

**Fig. 4 | Synthesis of additional C$_{16}$ building blocks in yeast.** Chromatograms of extracts of AM109 yeast cells expressing different *Sp*SodMT variants (blue) or empty vector (black) combined with bar charts showing the selectivity of *Sp*SodMT variants for the synthesis of different C$_{16}$ building blocks. Peaks marked with * correspond to indole-containing compounds from yeast. **a** Chromatogram of extract of wild-type *Sp*SodMT. **b** Chromatogram of extract of *Sp*SodMT(Q57N)-expressing yeast cells revealing production of **1a** and **1b** as conversion products of PSPP and **2a**, **2b** and **2d** derived from PPP. Minor amounts of **4b** were also observed (shown here in the inset by selecting for *m/z* 135). **c** Chromatogram of extract of *Sp*SodMT(F58V)-expressing yeast showing the synthesis of four C$_{16}$ compounds, **2a** and **2b** from conversion of PPP, and **3a** and **3b** from TPP. **d** Product profile of *Sp*SodMT(F58L)-expressing yeast cells. **e** Yeast cells expressing variant *Sp*SodMT(V273A). **f** Product profile of *Sp*SodMT(F58M-L302Q)-expressing cells. The peak indicated with ** corresponds to an uncharacterized compound possibly derived from KPP, as suggested by the observation that it is present only in samples where **6a** is also present. The area of this peak was not included in the calculation of the product profile of the different *Sp*SodMT mutants. **g** *Sp*SodMT(F58M-L302S)-expressing yeast cells. Samples were analyzed in triplicate (*n* = 3 biological replicates) and the mean value of the sum of products corresponding to each C$_{16}$ diphosphate was then determined and used to calculate the percentage of each product relative to the total C$_{16}$ products, shown here. Source data are provided as a Source Data file.

oxidoreductases, such as Oye2p[27,28] (Supplementary Table 1, Supplementary Figs. 1 and 2, Supplementary Note 1, Supplementary Data 2). Production of **1a** and **1b** in the AM109-based yeast system reached considerable titers of 15.3 and 10.7 mg L$^{-1}$, respectively, while the titers of the FPP-derived alcohols nerolidol and farnesol were 3.6 and 0.42 mg L$^{-1}$, respectively. As the levels of terpene alcohols produced by yeast cells under these conditions can be used as proxy of the intracellular levels of the corresponding prenyl diphosphates[23,24,26], these findings suggest that methylation of FPP in our yeast system is efficient, resulting in significant accumulation of the corresponding diphosphate, PSPP, at the expense of FPP levels.

We investigated whether PSPP biosynthesis interferes with the yeast's endogenous isoprenoid metabolism. First, we analyzed the growth characteristics of *Sp*SodMT-expressing cells and found no changes compared to the empty vector-containing control (Supplementary Fig. 3). We also confirmed that the sterol and squalene profile of cells expressing *Sp*SodMT was very similar to cells transformed with the empty vector, indicating that PSPP does not undergo head-to-head condensation by Erg9p and, thus, does not interfere with the sterol biosynthesis pathway (Supplementary Fig. 4). Furthermore, we analyzed the substrate specificity of *Sp*SodMT using in vitro assays with recombinant protein produced in *Escherichia coli* and substrates of different size (GPP, FPP and GGPP). Acid hydrolysis of the products of the reaction of *Sp*SodMT with FPP and *S*-adenosyl-methionine (SAM) resulted in the formation of **1b** (Supplementary Fig. 5), confirming synthesis of PSPP. However, no additional products were synthesized when GPP or GGPP were used as substrates, suggesting that it is unlikely that *Sp*SodMT methylates other prenyl diphosphates than FPP in the yeast cells. Overall, we concluded that the engineered FPP methylation pathway is orthogonal to the yeast's native metabolism.

**Engineering the synthesis of additional C$_{16}$ building blocks**

Having established synthesis of PSPP, we explored the possibility to develop additional non-canonical building blocks in order to create a larger alphabet for re-writing the terpene code. To this end, we examined whether *Sp*SodMT can be engineered to produce different C$_{16}$ prenyl diphosphates. The mechanism of PSPP synthesis by *Sp*SodMT involves a methyl group addition event leading to the formation of a carbocation, which, in turn, isomerizes to different forms before it is quenched by, possibly, a nearby residue (Fig. 3a and ref. 19). Thus, *Sp*SodMT can be considered as a bifunctional enzyme that combines, in the same active site, the function of a methyltransferase with that of a terpene synthase. We reasoned that it should be feasible to tamper with the terpene synthase reaction to tease out different products. We postulated that if the carbocation cascade that follows the methylation step could be altered by changes in the contour of the active-site pocket or in the physicochemical properties of the residues that line it, then this would result in the synthesis of structurally different prenyl diphosphate products. Thus, we constructed a model of *Sp*SodMT based on the coordinates of the structurally homologous norcoclaurine-6-*O*-methyltransferase from *Thalictrum flavum*[29] (PDB ID: 5kok) and docked the substrate, FPP, in this model (Fig. 3b). We identified 10 residues (F56, Q57, F58, S190, H191, N219, D272, V273,

W276, and L302) that line the FPP binding pocket, and designed rational substitutions in these residues that could derail the carbocation substrate, either by potential stabilization/destabilization of carbocationic intermediates, or by steric interference. In total, we constructed a library of 55 different *Sp*SodMT single-residue variants (Supplementary Data 3).

By screening the *Sp*SodMT mutant library in yeast, we identified several variants producing a different product profile than the parental enzyme, suggesting that they were able to synthesize alternative C$_{16}$ diphosphates (Fig. 4 and Supplementary Table 2, Supplementary Fig. 6). Among them, variants *Sp*SodMT(Q57N) and *Sp*SodMT(N219S) showed a very similar product profile characterized by the formation of two additional peaks (**2a** and **2b**) and a significant decrease in PSPP-derived alcohols compared to the parental enzyme (Fig. 4b and Supplementary Table 2, Supplementary Fig. 6b). We were able to isolate compound **2a** and confirm by NMR analysis that it has a distinct structure, shown in Fig. 1 (Supplementary Table 1, Supplementary Fig. 1, Supplementary Note 1, Supplementary Data 2). We named compound **2a** plymuthenol. We also purified and determined the structures of the tertiary alcohol **2b**, which we named plymutheloool, and **2c**, which corresponds to the 2,3-reduced form of **2a**, which we named 2,3-dihydro-plymutheloool (Supplementary Table 1, Supplementary Fig. 1, Supplementary Note 1, Supplementary Data 2). By analogy to the formation of **1a**, **1b**, and **1c**, by acid hydrolysis or yeast phosphatase-based hydrolysis of PSPP, we concluded that **2a**, **2b**, and **2c** are derived from a C$_{16}$ building block that we named plymuthenyl diphosphate (PPP; **2**) (Supplementary Table 1).

Substitution of F58 of *Sp*SodMT enabled the production of several more C$_{16}$ building blocks. We identified four variants of this residue (V, L, M and T) that, instead of PSPP, produced other main products with varying selectivity. Among these variants, *Sp*SodMT(F58V) became a weak but dedicated PPP synthase, producing 77.1% of this building block in yeast (Fig. 4c and Supplementary Table 2). *Sp*SodMT(F58L) and *Sp*SodMT(F58M), on the other hand, were rather promiscuous and synthesized several different C$_{16}$ building blocks, as deduced by the distinctive mass-spectral patterns of their hydrolysis products (Fig. 4d and Supplementary Table 2, Supplementary Figs. 6c and 7). To determine the chemical structure of the synthesized prenyl diphosphates, we selected to produce large-scale cultures of *Sp*SodMT(F58L)- and *Sp*SodMT(F58M)-expressing AM109 cells because of the broad product diversity achieved by cells expressing these enzyme variants. We were able to purify compounds **3a**, **3b**, **3c**, and **4a**, **4b**, and **4c**, which were found to have characteristic structures by analysis of their NMR data (Supplementary Table 1, Supplementary Fig. 1, Supplementary Note 1, Supplementary Data 2). The structures of these compounds revealed the synthesis of two diphosphate building blocks, **3** and **4**, shown in Figs. 1 and 3, which we name thorvaldsenyl diphosphate (TPP; **3**) and weylandtenyl diphosphate (WPP; **4**) (Supplementary Table 1).

Substitution of V273 of *Sp*SodMT with A, N, S or T resulted in biocatalysts that primarily synthesize WPP. In particular, yeast cells expressing variant *Sp*SodMT(V273A) produced WPP-derived

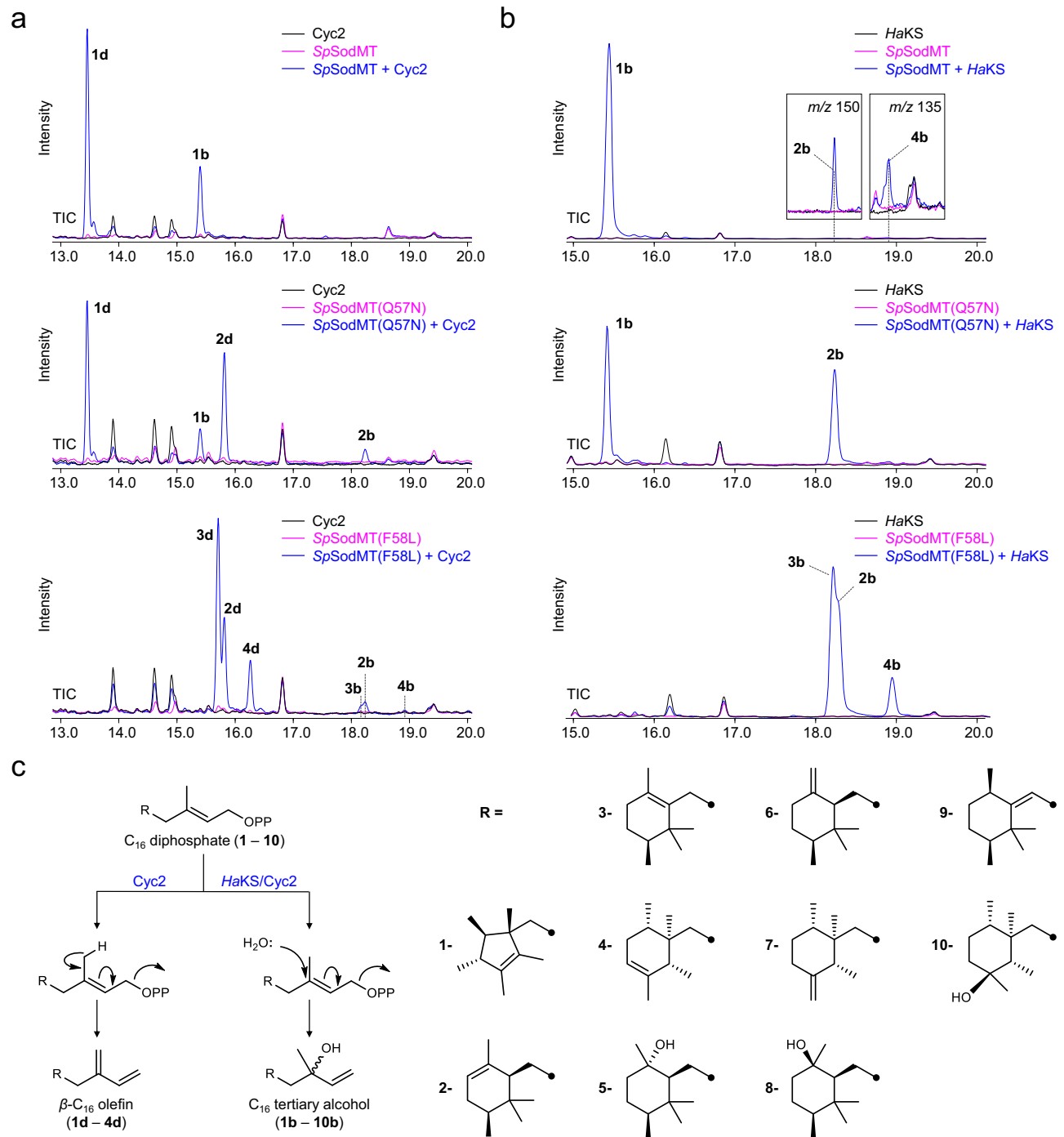

**Fig. 5 | Production of terpene scaffolds using the $C_{16}$ building blocks.** Chromatograms showing production of $C_{16}$ terpenes in yeast by expression of Cyc2 or *Ha*KS together with *Sp*SodMT wild-type or variants (blue) versus cells expressing the corresponding methyltransferase but lacking the terpene synthase (pink) or transformed with empty vector (black). **a** In yeast cells expressing *Sp*SodMT(Q57N), Cyc2 produced the $C_{16}$ olefin **1d** and the tertiary alcohol **1b** from PSPP, and **2d** and **2b** from PPP. Yeast cells expressing Cyc2 with *Sp*SodMT(F58L) (blue) showed production of **2d** and two $C_{16}$ olefins (**3d** and **4d**). **b** In cells expressing wild-type *Sp*SodMT, *Ha*KS synthesized pre-sodorifelool (**1b**) from PSPP (blue). Minor amounts of **2b** and **4b** were also observed (shown in the inset by selecting for *m/z* 150 or 135, respectively). Cells expressing *Sp*SodMT alone or *Ha*KS alone are shown as control (pink or black, respectively). In cells expressing *Sp*SodMT(Q57N), *Ha*KS produced **1b**

and **2b** from PSPP and PPP, respectively. A blend of tertiary alcohols, including **2b**, **3b**, and **4b**, were produced in cells expressing the promiscuous *Sp*SodMT(F58L) variant. In both panels **a** and **b**, yeast cells were cultivated in buffered media to prevent hydrolysis of excess prenyl diphosphates in acidified media during cultivation. **c** Proposed mechanism of Cyc2 and *Ha*KS reactions using $C_{16}$ diphosphates. In a reaction that involves an ionization-initiated cyclization cascade followed by deprotonation, Cyc2 converts the non-canonical building blocks to the corresponding $C_{16}$ olefins. *Ha*KS, and to a lesser extent Cyc2, produce non-canonical tertiary alcohols by addition of water to $C_{16}$ tertiary carbocation intermediates. Changes in the retention times of specific compounds in Fig. 5 as compared to those in Fig. 4 are due to different temperature program (GC analysis method 2, used in Fig. 5 vs. GC analysis method 1, used in Fig. 4) and instrument used in the current analysis.

compounds amounting to 80.3% of their total $C_{16}$ product profile (Fig. 4e and Supplementary Table 2).

Three additional building blocks, namely **5**, **6**, and **7**, were produced by L302 variants of *Sp*SodMT (Supplementary Table 2, Supplementary Fig. 6). The synthesis of these building blocks was evident by the isolation and structural characterization of compounds **5a**, **5b**, *epi*−**5b**, **6a**, **7a**, **7b** and **7c** (Supplementary Table 1, Supplementary Fig. 1, Supplementary Note 1, Supplementary Data 2). The corresponding prenyl diphosphates were named blixenyl diphosphate (BPP; **5**), kimlarsenyl diphosphate (KPP; **6**), and serratinyl diphosphate (SPP; **7**) (Supplementary Table 1). In *Sp*SodMT(L302Q)-expressing yeast cells, BPP (**5**) reached 40.8% of total $C_{16}$ (Supplementary Table 2, Supplementary Fig. 6d). Following identification of these six building blocks, we were able to confirm that BPP (**5**) was also produced in low amounts by *Sp*SodMT(F58M), KPP (**6**) was also made by *Sp*SodMT(F58L), while BPP (**5**) and KPP (**6**) were minor products of *Sp*SodMT(V273A) (Fig. 4d, e and Supplementary Table 2, Supplementary Fig. 6c). Moreover, WPP was also produced, albeit in low amounts, by SpSodMT(Q57N) (Fig. 4b and Supplementary Table 2).

To engineer additional synthases, we undertook a second round of site-directed mutagenesis. In this round, we focused on four residues (Q57, F58, V273 and L302) that were found to have the strongest impact on the product profile. By combining substitutions of a pair of these residues at a time, we constructed 15 double variants, which we screened in strain AM109 (Supplementary Data 1, Supplementary Table 3, Supplementary Fig. 8). This effort yielded promiscuous synthases *Sp*SodMT(F58M-L302S), *Sp*SodMT(F58M-L302Q), and *Sp*SodMT(F58M-V273A) (Fig. 4f, g and Supplementary Table 2, Supplementary Fig. 6). Isolation and characterization of compounds **8b**, *epi*−**8b**, **9b** and **10b**, produced by yeast cells expressing variants *Sp*SodMT(F58M-L302S), *Sp*SodMT(F58M-L302Q), or *Sp*SodMT(F58M-V273A), revealed the synthesis of three additional building blocks, jacobsenyl diphosphate (JPP; **8**), hammershoyl diphosphate (HPP; **9**), and ancheryl diphosphate (APP; **10**) (Supplementary Table 1, Supplementary Fig. 1, Supplementary Note 1, Supplementary Data 2).

In summary, the rational approach applied here resulted in the production of nine additional $C_{16}$ building blocks and the engineering of bio-catalysts that preferentially produce certain of these building blocks in yeast cells. These enzymes include one variant that produces mainly WPP (*Sp*SodMT(V273A)), one enzyme that favors production of TPP (*Sp*SodMT(F58L)), and one, albeit weak, synthase that synthesizes mostly PPP (*Sp*SodMT(F58V)) in yeast. In vitro experiments using FPP and SAM as co-substrates revealed that *Sp*SodMT(V273A) is indeed a selective WPP synthase that produces 75.7% WPP, and *Sp*SodMT(F58L) synthesizes TPP with 61.0% preference (Supplementary Table 4, Supplementary Fig. 9). We could not confirm the apparent selectivity of *Sp*SodMT(F58V) for PPP (Fig. 4c) using in vitro assays because this variant proved very inefficient or unstable. Also, because of the instability of certain products of BPP, KPP and JPP, we were not able to confidently determine the product specificity of *Sp*SodMT(F58M-L302Q) and *Sp*SodMT(F58M-L302S) using in vitro assays.

This strategy also led to several efficient enzymes and many variants supported $C_{16}$ product titers equal to or higher than the wild-type enzyme (Supplementary Fig. 10). For example, *Sp*SodMT(F58M) produced 34.4 mg L$^{-1}$ total $C_{16}$ compounds, while *Sp*SodMT(V273A) reached a total $C_{16}$ titer of 43 mg L$^{-1}$. Having identified key residues involved in product selectivity in *Sp*SodMT, further protein engineering can lead to the development of FPP methyltransferases with higher or different selectivity for non-canonical building blocks, enabling further expansion of the terpene chemical space.

## Using the newly identified building blocks to make non-canonical terpenes

Achieving production of these additional $C_{16}$ diphosphates in yeast establishes a platform for the synthesis of non-canonical terpenes.

Identifying or engineering terpene synthases that are able to convert the alternative building blocks into terpene scaffolds, followed by decoration of these structures by terpene modifying enzymes, will lead in the reconstruction of the entire modular structure of terpene biosynthesis on alternative precursors. As a first step towards this goal, we examined whether terpene synthases acting on FPP, the canonical $C_{15}$ prenyl diphosphate, could also accept $C_{16}$ diphosphates as substrates. We selected sesquiterpene synthases with varying product specificity, including both promiscuous[30–32] and specific[21,33] enzymes (Supplementary Table 5), and evaluated their ability to accept non-canonical substrates using yeast cells expressing *Sp*SodMT variants selected to cover all $C_{16}$ building blocks. However, the sesquiterpene synthases tested were not able to convert any of the $C_{16}$ diphosphates to products.

The inability of the terpene synthases tested to accept the alternative substrates may be due to the fact that their active site is designed to accommodate linear prenyl diphosphates and may, therefore, not be suitable to accommodate the penta- or hexacyclic ring of $C_{16}$ diphosphates. Thus, we reasoned that diterpene synthases involved in labdane-type terpene biosynthesis, enzymes that usually utilize cyclic prenyl diphosphates with 20 carbon atoms as substrates, may be more suitable to accommodate and convert a smaller monocyclic substrate, such as the building blocks developed here. Initially, we tested several abietane skeleton-producing enzymes (Supplementary Table 5) but again failed to obtain $C_{16}$ products. Subsequently, we turned into another subclass of labdane-type enzymes, the one producing terpene scaffolds of the clerodane type, and selected two enzymes with different catalytic mechanisms that have previously been shown to exhibit promiscuity with substrates of various lengths[34–37]. These included terpentetriene synthase (Cyc2) from *Kitasatospora griseola*, which converts terpentedienyl diphosphate to the bicyclic clerodane olefin terpentetriene[35], and *Herpetosiphon aurantiacus* kolavelool synthase (*Ha*KS)[38], which catalyzes the conversion of the bicyclic kolavenyl diphosphate to the tertiary alcohol (+)-kolavelool[37]. We found that both of these clerodane synthases were able to accept the various $C_{16}$ building blocks and synthesize non-canonical terpenes. When Cyc2 was expressed in PSPP-producing cells, we detected one major product (**1d**) and one additional peak corresponding to presodorifelool (**1b**) (Fig. 5a). Compound **1d** was readily purified from a large-scale culture of Cyc2 and *Sp*SodMT-expressing yeast cells, and characterized by NMR analysis as the PSPP-derived olefin β-presodorifene (Supplementary Table 1, Supplementary Fig. 1, Supplementary Note 1, Supplementary Data 2). The structure of the main product of Cyc2 with PSPP is consistent with the main product of the Cyc2-catalyzed conversion of FPP, (E)-β-farnesene[35,36]. Presodorifelool, on the other hand, is likely produced by addition of $H_2O$ to the tertiary allylic carbocation produced by Cyc2-mediated removal of the diphosphate group of PSPP, probably as a result of relaxed specificity of Cyc2 with the smaller $C_{16}$ substrate (Fig. 5c).

When Cyc2 was expressed in cells producing the other nine building blocks, we detected additional products (**2d**, **3d**, **4d**; Fig. 5b and Supplementary Fig. 11) with three of these substrates, PPP, TPP, and WPP. HRMS analysis of these products was consistent with a molecular formula of $C_{16}H_{26}$ (Supplementary Fig. 12). Therefore, given that the prototypical olefin-generating Cyc2 reaction is restricted to the isoprenyl unit proximal to the diphosphate group, we presume that, like **1d**, compounds **2d**, **3d**, and **4d** are also olefins derived from the corresponding building blocks following diphosphate removal and formation of an additional carbon-carbon double bond without alteration of the core ring structure. To obtain a better understanding of the identity of these products, we extracted a mixture of **1d** and **2d** from a large-scale culture of AM109 cells co-expressing Cyc2 and *Sp*SodMT(Q57N). Following analysis of its NMR data, we elucidated the structure of compound **2d** (Supplementary Table 1, Supplementary Fig. 1, Supplementary Note 1, Supplementary Data 2), confirming the

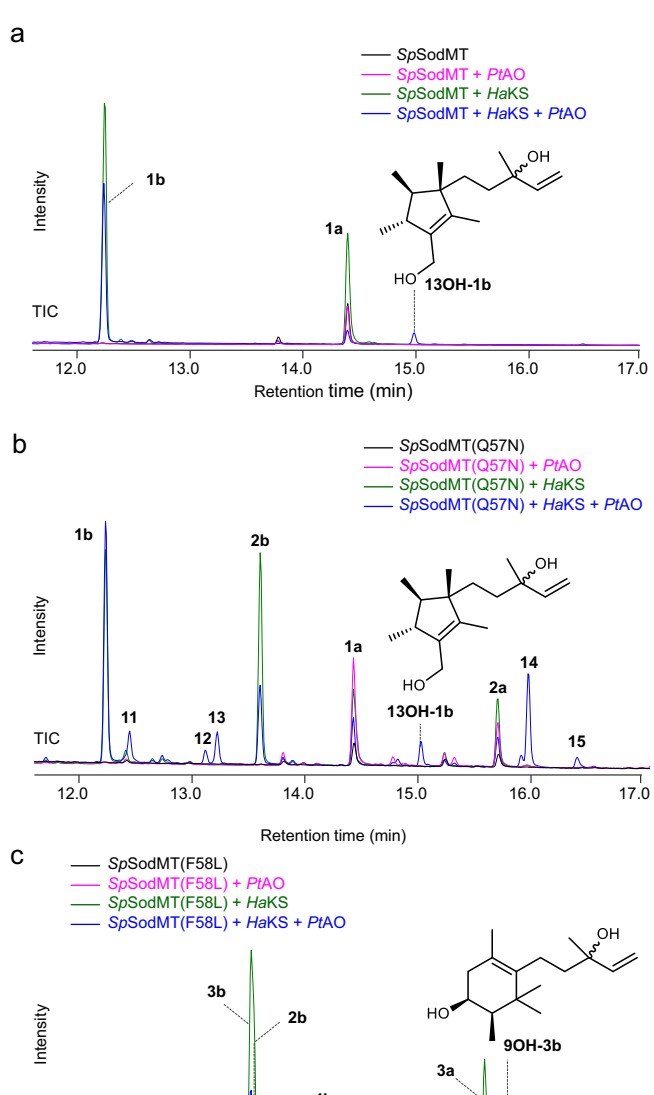

**Fig. 6 | Production of complex C$_{16}$ terpenes by CYP720B1.** Chromatograms showing the product profile of CYP720B1 (*Pt*AO) in yeast cells co-expressing *Ha*KS with *Sp*SodMT wild-type or variants (blue) versus control cells transformed with *Sp*SodMT wild-type or variants but lacking *Ha*KS or *Pt*AO or both. **a** In PSPP-producing yeast cells, *Pt*AO oxidized **1b** to produce **13OH-1b**. **b** When, both PSPP- and PPP-derived C$_{16}$ alcohols were produced by yeast cells, *Pt*AO produced **13OH-1b** and compounds **11-15**. **c** In cells expressing *Sp*SodMT(F58L), *Pt*AO produced **9OH-3b** from **3b**.

hypothesis that it corresponded to the PPP-derived olefin. Compound **2d** was named β-plymuthene. Although we were not able to isolate in pure form **3d** and **4d** in order to verify their structures by NMR data, based on the structural characterization of **1d** and **2d**, it is likely that their structures also conform to the β- configuration of the corresponding building block-derived olefins. In addition to these products, Cyc2 also produced lower amounts of compounds **2b-4b** by water addition to the tertiary allylic carbocation of building blocks **2-4**, as was the case with **1** (Fig. 5a). We further confirmed that Cyc2 is able to convert the C$_{16}$ prenyl diphosphates by carrying out in vitro reactions with *Sp*SodMT variants and Cyc2 using FPP and SAM as substrates (Supplementary Fig. 13).

When *Ha*KS was expressed in cells producing different C$_{16}$ building blocks, we observed specific production of the corresponding C$_{16}$ tertiary alcohols (Fig. 5b and Supplementary Table 6). This is in agreement with the *Ha*KS reaction mechanism, which involves nucleophilic attack by a water molecule to the tertiary position of the allylic carbocation produced upon cleavage of the diphosphate group (Fig. 5c). The ability of *Ha*KS to convert the C$_{16}$ substrates was confirmed by in vitro reactions using *Sp*SodMT variants and FPP and SAM as co-substrates (Supplementary Fig. 14). Notably, we were able to detect minor amounts (less than 1%) of **2b** and **4b** in the extracts of yeast cells expressing *Sp*SodMT together with *Ha*KS (Fig. 5b), suggesting that, at least when produced in yeast, wild-type *Sp*SodMT likely also synthesizes building blocks **2** and **4**. At this stage, it is unclear whether *Sp*SodMT also synthesizes minor amounts of **2** and **4** in *S. plymuthica*.

The product titers obtained in shake flask cultivation were considerable, allowing further expansion of the pathway. *Ha*KS supported the production of **1b** at 59.3 mg L$^{-1}$ (Supplementary Table 6). In the case of Cyc2, product titers of **1d** and **2d** were 11.6 mg L$^{-1}$ and 5.4 mg L$^{-1}$, respectively (Supplementary Table 7).

## Decoration of the non-canonical terpene scaffolds by P450 oxidation

To further expand the diversity of chemical structures created, we proceeded to identify enzymes able to decorate the non-canonical C$_{16}$ terpene scaffolds produced. Terpene oxidation, typically carried out by cytochrome P450 enzymes (CYPs), is a frequently encountered modification of terpene scaffolds. Thus, we set out to produce complex non-canonical terpenes by introducing downstream CYP-driven oxidation steps in the established C$_{16}$ biosynthetic pathways. To this end, we selected 9 CYPs with varied substrate selectivity and product specificity (Supplementary Table 5) and introduced them, together with a compatible redox partner (CPR2 from *Poplar x media*[39]), into yeast strains co-expressing *Sp*SodMT variants and *Ha*KS (because of its higher efficiency). Among these, we found that CYP720B1 (*Pt*AO) was able to oxidize C$_{16}$ scaffolds, producing several products derived from different building blocks. Specifically, in PSPP producing yeast strains, an additional C$_{16}$ product was detected (Fig. 6a, b). Isolation and NMR analysis revealed the structure of this compound to be 13-hydroxy-presodorifelool (**13OH-1b**) (Supplementary Table 1, Supplementary Fig. 1, Supplementary Note 1, Supplementary Data 2), confirming that it is the product of PtAO-mediated hydroxylation at C-13 of **1b**. Titers of **13OH-1b** reached 35.34 mg L$^{-1}$ (Supplementary Table 8).

We also observed the synthesis of five tentative C$_{16}$ PtAO-mediated oxidation products (**11-15**) in cells expressing *Sp*SodMT(Q57N) (Fig. 6b and Supplementary Fig. 15) but since they were not isolated in pure form, their structures were not elucidated. Instead, we focused on *Sp*SodMT(F58L)-expressing cells and were able to isolate one additional PtAO product and elucidate its structure to be 9-hydroxy-thorvalselool (**9OH-3b**) (Supplementary Table 1, Supplementary Fig. 1, Supplementary Note 1, Supplementary Data 2), further confirming the oxidation of non-canonical C$_{16}$ terpenoids by PtAO. The product titer of **9OH-3b** reached 8.82 mg L$^{-1}$ (Fig. 6c and Supplementary Table 8).

## Olfactory properties of the produced non-canonical terpenes

Mono- and sesquiterpenes frequently have pleasant sensory properties and have found broad application as fragrant compounds and flavoring agents. To investigate whether the compounds produced using this approach also possess interesting sensory characteristics, we undertook a proof-of-concept study using GC-MS-olfactometry (GC-MS-O) focusing on the purified C$_{16}$ terpenes. Initial evaluation of the pure compounds by a panel of three trained odor-assessors selected a subset of nine odor-active compounds among the 28 isolated C$_{16}$ molecules. These included **13OH-1b**, **3a**, **9OH-3b**, **4a**, **4b**, **5b**,

**Table 1 | Odor descriptors of main odor-active C$_{16}$ terpenes**

| Compound | Odor descriptor |
|---|---|
| **3a** | burnt |
| **9OH-3b** | burnt rose |
| **4a** | old beer |
| **epi−5b** | woody |

**epi−5b**, **7b**, and **9b**. To eliminate possible interference from minor impurities or degradation products present in the samples, the nine shortlisted compounds were further evaluated by GC-MS-O. Four of these, **3a**, **9OH-3b**, **4a**, and **epi−5b**, were determined to have characteristic odors with low to medium odor thresholds and the descriptors shown in Table 1. These results open up the possibility to identify valuable fragrant compounds using synthetic biology approaches such as the one presented in this study.

## Discussion

Synthetic biology aims to re-design existing biological systems to acquire different functions for useful purposes. In this context, engineering cells to synthesize and utilize non-canonical building blocks for key metabolic pathways will enable the synthesis of compounds with structures not currently found in biological systems. This will facilitate the exploration of uncharted areas of the chemical space and provide molecules with improved properties for industrial applications.

Isoprenoids hold a tremendous potential for engineering synthetic biochemical pathways leading to non-canonical compounds. This is because isoprenoid biosynthesis has a highly consistent modular structure that dramatically expands the number and complexity of molecules synthesized from each building block[17]. Thus, introduction of just one non-canonical building block can lead to a plethora of additional compounds. Previous efforts to expand isoprenoid biosynthesis by incorporating non-canonical C$_6$ precursors showed that many enzymes involved in isoprenoid pathways exhibit sufficient promiscuity to allow incorporation of the non-canonical blocks into larger size products[40,41,42]. However, two key challenges limited the effectiveness of these efforts in achieving the production of specific isoprenoids. First, the non-canonical C$_6$ precursors were indiscriminately directed to several different branches of isoprenoid biosynthesis in the host cells rather than to a specific compound class[41]. Secondly, the carbon-size and overall structure of the resulting products was not fixed but consisted of a mixture of different sizes and structures depending on the combination of canonical and non-canonical building blocks used for their construction[41]. Moreover, introducing non-canonical building blocks in the early steps of isoprenoid biosynthesis increases the risk for undesirable interactions with the host metabolism that may compromise fitness of the host organism. Therefore, to achieve the efficient and systematic production of non-canonical isoprenoid compounds, it is essential to engineer cells to synthesize class-specific prenyl diphosphate building blocks, which, at the same time, are orthogonal to the host metabolism. In this case, introducing the subsequent modules of terpene biosynthesis, such as P450-driven oxidations, reductions, or acetylations, will enable the reconstruction of complete non-canonical terpenoid pathways. By optimizing the enzymes and pathways involved, this approach can develop efficient cell factories specific for the production of compounds with desirable properties and eventually lead to their sustainable industrial production.

In this work, we specifically address this challenge. We successfully apply protein engineering to enable yeast to synthesize a collection of C$_{16}$ prenyl diphosphates not previously observed in nature, and confirm that these building blocks are orthogonal to the yeast isoprenoid pathways. We subsequently use these alternative terpene building blocks to reconstruct successive steps (modules) of terpene biosynthesis by identifying terpene synthases to construct non-canonical terpene scaffolds and a cytochrome P450 enzyme to decorate the terpene structures. In total, we produce at least 28 compounds with different structures and varying degrees of oxygenation that have not been observed in a biological system before.

Our findings offer an unprecedented example of the use of protein engineering of a methyltransferase as a tool to create different terpene scaffolds. This is based on the characteristic mechanism of SpSodMT that combines the activity of a methyltransferase, which introduces a methyl group at C-10 of FPP, with that of a terpene synthase, which engages the methylation reaction-activated linear prenyl diphosphate in a carbocationic cascade that leads to the formation of a cyclic prenyl diphosphate. Based on previous understanding of terpene synthase catalytic selectivity[32,40,43,44], we were able to alter the cyclization mechanism and engineer SpSodMT to produce nine alternative building blocks. Further protein engineering can expand the diversity created.

Immediate applications of the produced non-canonical C$_{16}$ terpenes can be envisioned in the area of flavors and fragrances. Our preliminary olfactory analysis indicated that several C$_{16}$ terpenes possess characteristic odor properties. Comprehensive sensory evaluation of the produced compounds combined with the efficient and sustainable yeast production platform established here, may lead to different natural and organic cosmetics, or food and beverage ingredients.

Our study also raises important questions about the biological relevance of the synthesized non-canonical C$_{16}$ terpenes. Are these molecules part of a chemical diversity never synthesized and tested by living organisms, or are some of these compounds components of undiscovered biological systems? Given that sodorifen has been implicated in the interaction of S. plymuthica with the fungus Fusarium culmorum[45], it would be interesting to introduce the pathways developed here in bacteria and evaluate how this affects their interaction with plants or other microorganisms.

## Methods

### Chemicals and enzymes

Standards used include: trans-nerolidol (Sigma-Aldrich, 04610590), farnesol (Sigma-Aldrich, F203), linalool (Aldrich, L2602). All C$_{16}$ compounds isolated from yeast cultures and characterized by NMR were used as in-house standards. Substrates used for in vitro reactions of recombinant SpSodMT wt protein with SAM (A2408-Sigma) and geranyl diphosphate (G6772; Sigma-Aldrich), farnesyl diphosphate (F6892; Sigma-Aldrich), and geranylgeranyl diphosphate (G6025; Sigma-Aldrich). Phusion High-Fidelity DNA Polymerase (New England BioLabs, M0530S) and MyTaq DNA polymerase (BIO-21105, Bioline) were used in PCR amplifications. QIAquick Gel Extraction Kit (#28704, Qiagen) was used for gel extraction and DNA purification. NucleoSpin Plasmid Kit (740588.250, Macherey-Nagel) was used for plasmid DNA purification.

### Yeast media

D-( + )-glucose monohydrate (16301, Sigma); D-( + )-galactose (G0625, Sigma); raffinose pentahydrate (R1030, US Biological); Yeast Nitrogen Base w/o AA (Y2025, US Biologicals); Complete Minimal (CM) medium was composed of 0.13% (w/v) dropout powder (all essential amino acids), 0.67% (w/v) yeast nitrogen base w/o AA, 2% glucose; For galactose based medium, glucose was substituted with 2% galactose, 1% raffinose. Buffered media contained 200 mM MES (M2933, Sigma) and the pH adjusted to 6.5.

### Gene cloning and expression in yeast

All primers used for cloning purposes are listed in the Supplementary Table 2. Yeast codon optimized versions of S. plymuthica sodorifen

methyltransferase (*Sp*SodMT) (GenBank WP_004943917.1), *A. grandis* γ-humulene synthase (*Ab*HumS; GenBank U92267), *K. griseola* terpentetriene synthase (Cyc2; GenBank AB048795.1), and *H. aurantiacus* kolavelool synthase (*Ha*KS, GenBank ABX04786) were obtained by gene synthesis bearing flanking regions containing specific restriction sites and a generic sequence compatible for USER cloning. Using the generic primes USER-Gen-FP and USER-Gen-RP, the above genes were amplified and cloned by USER cloning into a generic backbone introducing *Bam*HI and *Eco*RI sites at the 5′ end, and *Not*I, *Bgl*II and *Xho*I sites at the 3′ end. The constructs were confirmed by sequencing. Subsequently, the genes of interest were excised from the generic USER vector by *Bam*HI and *Xho*I digestion and ligated into pESC vector (Agilent Technologies, Cat. #217453) of different selection (TRP for *Sp*SodMT, HIS for TPSs) linearized with *Bam*HI and *Xho*I restriction enzymes. This approach enabled subcloning of the genes of interest under P$_{GAL1}$ promoter and resulted into construction of the following plasmids: pESC-TRP/*Sp*SodMT, pESC-HIS/*Ab*HumS, pESC-HIS/CYC2, and pESC-HIS/*Ha*KS. Genes encoding the different other TPSs involved in this study (full list available in Supplementary Table 5) were available from previous work as compatible parts for subcloning in pESC-HIS linearized with *Bam*HI and *Xho*I[21,31–33].

pESC-Leu/CPR2 vector carrying the cDNA encoding a cytochrome P450 reductase from poplar (CPR2)[39] was previously modified[46] to enable standardized CYPs cloning under the P$_{GAL10}$ promoter into the *Eco*RI and *Not*I restriction sites. Yeast codon optimized versions of CYP71AV1 (GenBank DQ318192), CYP706M1 (GenBank JX518290) and CYP76AK1 (GenBank KR140169) were obtained by gene synthesis carrying flanking regions containing *Eco*RI and *Not*I restriction sites. The rest of CYPs (full list available in Supplementary Table 5) were available from previous work[14,46,47]. All CYPs involved in this study were digested with *Eco*RI and *Not*I and subcloned in pESC-Leu/CPR2 linearized with the same restriction enzymes to obtain bifunctional constructs for simultaneous expression of one CYP and CPR2.

For protein expression in *E. coli*, a bacterial codon optimized version of *Sp*SodMT was obtained and cloned into pRSETa (Invitrogen) applying the USER technology described above, to generate the corresponding bacterial expression construct pRSETa/*Sp*SodMT. Selective mutations were introduced by USER mutagenesis[48].

Constructs pESC-HIS/*Pa*LAS and pESC-LEU/CPR2-*Pt*AO was kindly provided by Professor Joerg Bohlmann (University of British Columbia).

## Strain AM109
In *S. cerevisiae*, heterozygous deletions lead to 50% decrease in the level of the corresponding protein in the majority of yeast genes tested[49]. Strain AM109 (Mat a/α, P$_{GAL1}$-HMG2(K6R)::HO×2, ura3, trp1, his3, P$_{TDH3}$-HMG2(K6R)x2-::leu2, ERG9/erg9, UBC7/ubc7, SSM4/ssm4, PHO86/pho86), the basis strain used in this study, is a diploid strain that contains a heterozygous deletion in the squalene synthase gene ERG9, which is essential for sterol biosynthesis. This deletion helps reduce the drain of FPP substrate and boost the heterologous production of C$_{16}$ terpenoids[21].

## Small-scale C$_{16}$ production and quantification in yeast
Selected *S. cerevisiae* strains were cultivated in 10 mL CM liquid media. Liquid-liquid extraction of 1 mL culture using hexane was applied for measuring the C$_{16}$ compounds produced in yeast cells after 48 h of culturing followed by GC-MS and GC-qTOF analysis (see details in the corresponding section below). For quantification, 0.01% isopropyl myristate was used as internal standard. Samples were analyzed in triplicates and the error calculated as the mean absolute deviation (MAD) around the mean. Purified C$_{16}$ compounds obtained from yeast were used as standards when available to quantify the titer of specific compounds. Titers of compounds **1d** and **2d** were calculated using the corresponding standards, while the average signal peak area of **1d** and **2d** standards was calculated and used for quantification of the titers of

compounds **3d** and **4d**. Titers of compounds **1b-5b** and **7b-10b** were calculated using their corresponding standards. Titer of compounds **13OH-1b** and **9OH-3b** produced by *Pt*AO were calculated based on the purified **13OH-1b** and **9OH-3b** standards, respectively. The average signal of **13OH-1b** and **9OH-3b** was used for quantification of compounds **11-15**. Titers of total C$_{16}$ products were calculated using the average of standards corresponding to the most representative products in the profile of each individual mutants.

## Shake flask large scale cultivation
For the extraction of C$_{16}$ olefins, 1% (w/v) Diaion HP20 (Supelco, Bellefonte, PA) was used as adsorbent resin. The HP20 beads were activated by ethanol and washed by Milli-Q water before use. A two liter culture of yeast cells in Gal/Raff-CM medium together with 2 g HP20 beads were incubated in shake-flasks at 30 °C for 3 days. The medium and beads were subsequently extracted with hexane (3 × 1 L) at room temperature. The resulting hexane phase was evaporated to dryness by rotary evaporator.

## Fed-batch fermentation
Seed cultures were prepared by inoculating 5 mL of overnight cell culture into 500 mL selective CM glucose media and cultivated at 30 °C and 150 rpm. After 6 h of growth, the seed cultures were washed 3 times with water and added to a 5 L bioreactor (Sartorius benchtop bioreactor with BIOSTAT B Plus controller), containing two liter of CM galactose/raffinose media. Fermentation was carried out at 30 °C with the pH maintained at 4.2 by automatic addition of 4 M NaOH. Airflow was maintained at 1 L/min for the first 24 h, increased to 2 L/min from 24 h to 48 h and finally set to 10 L/min after 48 h for the remainder of the fermentation. The dissolved oxygen concentration was controlled at above 20% saturation by agitation cascade between 100-500 rpm. 200 mL of 15× concentrated media (30% galactose, 15% raffinose, 10% YNB, 2% dropout mix) was manually fed into the culture at 48 h, with subsequent daily feedings of 400 mL 15× media. Fermentations were terminated after 7–9 days and the cultures extracted.

## Protein expression in bacteria and enzymatic assays
*Sp*SodMT wild type, *Sp*SodMT(Q57N), *Sp*SodMT(F58L), and *Sp*SodMT(V273A) were expressed in *E. coli* BL21 [DE3, pLANT(3)/RIL[50]]. Briefly, bacterial cells carrying the corresponding plasmids were grown at 23 °C until OD$_{600}$ reached 0.5–0.7, then induced with 0.1 mM isopropylthio-β-galactoside (IPTG), after which cell growth was continued for 20 h at 19 °C. The His-tagged proteins were then purified by Ni$^{2+}$ affinity chromatography and eluted with 60 mM Tris-HCl (pH 8.8), 0.5 M NaCl, 250 mM imidazole, and 8% glycerol. The enzymatic activity of *Sp*SodMT was assayed in 0.2 mL reaction containing 10 mM HEPES-KOH (pH 8.0), 20 mM MgCl$_2$, 0.2 mM MnCl$_2$, 10 mM DTT and using as co-substrates 120 μM SAM and 70 μM of GPP, FPP or GGPP. The reactions were incubated for 18 h at 30 °C and terminated by addition of 0.2 mL 2 N HCl in 83% ethanol and 0.2 mL hexane. Acid hydrolysis of the diphosphates was carried for 20 min at 37 °C and the reactions were neutralized with 10% (w/v) NaOH and analyzed by GC-MS. All enzymatic assays were carried out in triplicates.

## In vitro assays using yeast-expressed *Sp*SodMT variants
*S. cerevisiae* strains containing plasmids for expression of *Sp*SodMT wild-type and mutants in combination with empty vector or terpene synthases Cyc2 or *Ha*KS, were cultivated in 50 mL of liquid media to a OD$_{600}$ of 1. Cells were harvested by centrifugation, washed once with water and resuspended in 2 mL Eppendorf tubes using 500 μL of 50 mM HEPES-KOH (pH 8.0) containing protease inhibitor (cOmplete™ Protease Inhibitor Cocktail, Sigma-Aldrich, 11697498001). 500 μL glass beads (G8772; Sigma-Aldrich) were added to each tube and the cells broken by alternating 1 min of vortexing and 1 min of cooling on ice for a total of 10 cycles. The broken cells centrifuged

21,000 xg for 5 min and the clarified lysate used as enzyme source. The enzymatic activity was assayed in 250 μL reactions containing 95 μL yeast lysate, 4 mM SAM, 100 μM FPP, 30 mM DTT, 20 mM MgCl₂, 0.5 mM MnCl₂, 10% (w/v) glycerol and 50 mM HEPES-KOH (pH 8.0). The reactions were incubated for 3 h at 30 °C and terminated by addition of 250 μL 2 N HCl in 83% ethanol and 125 μL hexane. Acid hydrolysis of the diphosphates was carried for 20 min at 37 °C and the reactions were neutralized with 10% (w/v) NaOH and the hexane analyzed by GC-MS analysis method 1 as specified below under subheading "GC analysis conditions".

### Yeast growth assay

The engineered yeast strains were grown in glucose-based media until $OD_{600}$ ~0.5–0.7, washed twice with sterile water and resuspended in galactose/raffinose-based media for induction of protein expression. The obtained cultures were diluted to $OD_{600}$ ~0.1 (time 0) in a final volume of 25 mL and were incubated at 30 °C with 150 rpm shaking. Measurements of $OD_{600}$ were taken every 2 h to monitor the curve of yeast growth (Supplementary Fig. 3). For $OD_{600}$ values higher than 1, samples were diluted accordingly.

### Analysis of yeast sterols and squalene

A volume of 4 mL yeast cultures induced to express SpSodMT (or containing the empty pESC-TRP vector) was used to collect yeast cells. The obtained pellet was treated with butylated hydroxytoluene (BHT) in a ratio of 2:1 (v/w), resuspended in 500 μL 10% KOH in 80% EtOH solution and incubated at 70 °C for 2 h. Samples were cooled at room temperature and overlaid with 300 μL of hexane followed by vigorous vortexing prior to collecting the hexane phase. The extraction was repeated three times and the collected hexane phase (~0.9 mL) was initially washed with water (equal volume) three times and then evaporated down to 100 μL and derivatized with 50 μL trimethylsilyl cyanide (Sigma, Cat. #212849) for 30 min at 60 °C prior to GC-MS analysis[46].

### GC analysis conditions

GC-MS analysis method 1 was performed on a Shimadzu GC-2010 system equipped with a GCMS-QP2010 Plus mass spectrometer, and was carried on a HP-5ms UI column (30 m x 0.25 mm ID x 0.25 μm film thickness) using helium as a carrier gas with a linear velocity of 30 cm/sec and the following temperature program: initial temperature 60 °C, ramp to 300 °C at a rate of 10 °C/min, hold for 3.5 min. This method was used for analysis of samples resulting from: incubation of the SPME fiber for 30 min over the head-space of 2-10 mL yeast cultures, 2 μL of the hexane phase overlaying the E. coli enzyme sourced purified in vitro enzymatic reactions, and 1 μL injections of hexane from the hexane extractions of non-saponifiable lipid and acid hydrolyzed in vitro enzymatic reactions using S. cerevisiae expressed SpSodMT wild-type and mutants.

GC-MS analysis method 2 was performed on a Shimadzu Nexis GC-2030 system equipped with a GCMS-QP2020 NX mass spectrometer and an HP-5ms column (30 m x 0.25 mm ID x 0.25 μm film thickness). He carrier gas was helium with a linear velocity of 30 cm/sec and the following temperature program was used: initial temperature 60 °C, ramp to 170 °C at 10 °C/min, ramp to 210 at 2 °C/min, ramp to 300 °C at 20 °C/min, hold for 5 min. This method was used for analysis of samples resulting from hexane extractions of S. cerevisiae grown in buffered media expressing SpSodMT together with Cyc2 or HaKS, and SPME fiber sampling of the head-space of in vitro enzymatic reactions using S. cerevisiae expressed SpSodMT together with Cyc2 or HaKS.

GC-APCI-QqToF experiments were performed on a Scion 456 GC system (Bruker Daltonics) with a micrOTOF-Q II (QqToF) mass spectrometer (Bruker Daltonics) using an atmospheric pressure chemical ionization source (APCI). Samples (1 μL) were injected at 250 °C in splitless mode using a BR-5ms capillary column (30 m x 0.25 mm ID

x 0.25 μm film thickness; Bruker Daltonics). The carrier gas was helium with a constant flow of 1.0 mL min⁻¹. The temperature program was: 50 °C for 1 min, ramp to 280 °C at 15 °C min⁻¹, and hold for 3.67 min. The transfer line was maintained at 280 °C and the temperature of the APCI source was 280 °C and of the APCI heater 200 °C. The charging voltage was 2000 V. The mass spectrometer was operated in full scan positive ion mode with the following settings: $m/z$ 80 − 310: nebulizer gas (nitrogen), 3.5 bar; drying gas (nitrogen), 2.5 L min⁻¹; drying gas temperature, 240 °C; capillary voltage, 3000 V; spectra acquisition rate, 5 Hz. Calibration for accurate mass was performed at the beginning of each chromatographic run using perfluorotributylamine vapour (PFTBA; Sigma Aldrich). Data acquisition was performed using Compass CDS (Version 3.0.1; Bruker Daltonics), Compass oTOF Control (Version 3.4, Bruker Daltonics) and Hystar (Version 3.2 SR4, Bruker Daltonics). Data analysis was performed using Compass DataAnalysis software (Version 4.3, Bruker Daltonics).

### GC-MS-olfactometry

Samples containing 100−300 μg of each of the isolated compounds were directly injected on Tenax-TA traps (200 mg of Tenax-TA, mesh size 60/80, Markes International, Llantrisant, UK). The trapped volatiles were thermally desorbed (TurboMatrix 300, Perkin Elmer, Shelton, USA) using a two-step procedure: Primary desorption was at 250 °C with a 50 mL min⁻¹ flow of carrier gas for 15.0 min. The stripped volatiles were trapped in a Tenax-TA cold trap (30 mg held at 5 °C), which was subsequently heated at 300 °C for 4 min (secondary desorption, outlet split 1:10). This allowed for rapid transfer of volatiles to a GC-MS (6890 N GC interfaced with a 5973 MSD from Agilent Technologies, Palo Alto, California) through a heated (225 °C) transfer line. Separation of volatiles was carried out on a ZB-Wax capillary column 30 m long x 0.25 mm internal diameter, 0.50 μm film thickness. The column pressure was held constant at 2.3 psi resulting in an initial flow rate of 1.4 mL min⁻¹ using hydrogen as carrier gas. The column temperature program was: 10 min at 30 °C, from 30 °C to 240 °C at 8 °C min⁻¹, and finally 5 min at 240 °C. The effluent of the column was split in a ratio of 1:2:2 between the mass spectrometer (electron ionisation mode at 70 eV, scanning $m/z$ ratios between 15 and 300) and two olfactory detector ports (ODP-2, Gerstel, Mülheim an der Ruhr). Agilent Chemstation (E.02.02.1431) was used for data acquisition and data analysis.

### Construction of the SpSodMT structural model

The structural model of SpSodMT was constructed using the SWISS-MODEL server[51] and 5kok[29] as template, scoring a GMQE of 0.52 and a QMEAN of −2.70 and covering amino acids 31-325. S-adenosylhomocysteine (SAH) was imported into the model from 5kok and FPP was chosen as a ligand from the ZINC database (ID: ZINC12494625). Using the SwissDock program, FPP was not found to dock next to SAH in the expected pocket. VegaZZ AMMP edition was used to optimize SpSodMT model by selecting the Conjugate Gradient (CG) algorithm for energy optimization and the CHARMM force field for fast docking. A total of 645 steps were run. The docking results using the optimized SpSodMT model demonstrated that FPP was placed next to SAH as expected. To obtain a more accurate model, the optimization process was repeated following conversion of SAH into S-adenosylmethionine (SAM). The final model displayed a deeper cavity for FPP than the initial model, and 4 Å distance from SAM to FPP, as expected for a methyl transfer[52,53]. Subsequently, a refined SpSodMT model was obtained using Molecular mechanics (MM2) algorithm[54]. The energy optimization was performed with the software Chem3D (v. 16.0) from CSC, Cambridge (USA). Residues F56, Q57, F58, G187, I189, S190, N219, V273, W276, L302, and W306 were identified within 5 Å of the isoprene unit tail, and were selected as potential mutagenesis targets for modifying product specificity.

## Mutagenesis

Site-directed mutagenesis of *Sp*SodMT was performed by USER mutagenesis[48] using the Uracil containing primers listed in Supplementary Data 3. In short, pESC-W/*Sp*SodMT plasmid was PCR amplified using PfuX7 DNA polymerase[55]. The resulted PCR products were digested by USER enzyme (New England BioLabs, M5505) and transformed into *E. coli*. Transformant colonies were cultured, miniprepped and confirmed by sequencing.

## General procedures for isolation and structure elucidation

Column chromatography separations were performed with Kieselgel 60 (Merck, Darmstadt, Germany). Normal phase HPLC separations were conducted using an Agilent 1100 liquid chromatography system equipped with refractive index detector (Agilent Technologies, Waldbronn, Germany) or a Cecil 1100 Series liquid chromatography pump (Cecil Instruments Ltd., Cambridge, UK) equipped with a GBC LC-1240 refractive index detector (GBC Scientific Equipment, Braeside, VIC, Australia), using an Econosphere Silica 10 u (250 × 10 mm, Grace, Columbia, MD, USA), Kromasil 100-10-SIL (250 × 10 mm, Akzonobel, Eka Chemicals AB, Separation Products, Bohus, Sweden), or Supelcosil SPLC-Si 5 μm (250 × 10 mm, Supelco, Bellefonte, PA, USA) column. Reversed phase HPLC was conducted on an Agilent 1200 HPLC system (Santa Clara, CA, USA) consisting of a G1367C high-performance autosampler, a G1311A quaternary pump, a G1322A degasser, a G1316A thermostated column compartment, a G1315C photodiode-array detector, and a G1364C fraction collector, all controlled by Agilent ChemStation version B.03.02 software. Separations were first performed at 40 °C on a semi-preparative Phenomenex Luna $C_{18}$(2) column (150 × 10 mm i.d., 5 μm particle size, 100 Å pore size; Phenomenex, Torrance, CA, USA) or a Phenomenex Luna $C_5$(2) column (150 × 4.6 mm i.d., 5 μm particle size, 100 Å pore size; Phenomenex, Torrance, CA, USA). Chiral HPLC separations were conducted on a Pharmacia LKB 2248 liquid chromatography pump (Pharmacia LKB Biotechnology, Uppsala, Sweden) equipped with an RI-102 Shodex refractive index detector (ECOM spol. s r.o., Prague, Czech Republic), using a Chiralcel OD, 10 μm (25 cm × 10 mm, Daicel Chemical Industries Ltd.) column. TLC were performed with Kieselgel 60 $F_{254}$ aluminum plates (Merck, Darmstadt, Germany) and spots were detected after spraying with 20% $H_2SO_4$ in MeOH reagent and heating at 100 °C for 1 min. NMR spectra were recorded on a Bruker DRX 400 spectrometer, a Bruker Avance III 600 instrument equipped with a Bruker SampleJet sample changer and a cryogenically cooled 1.7 mm TCI probe head for 2D NMR, a Bruker Avance III HD 600 spectrometer equipped with a cryogenically cooled 5 mm DCH probe optimized for ¹H and ¹³C (Bruker BioSpin GmbH, Rheinstetten, Germany) or a Varian 600 spectrometer. Chemical shifts are given on a $\delta$ (ppm) scale using TMS as internal standard. The 2D experiments (HSQC, HMBC, COSY, NOESY) were performed using standard Bruker or Varian pulse sequences. Low-resolution EI mass spectra were measured on a 5977B mass spectrometer (Agilent Technologies). High-resolution APCI mass spectra were recorded on a Scion 456 GC system (Bruker Daltonics) coupled to a micrOTOF-Q II (QqToF) mass spectrometer (Bruker Daltonics) via an atmospheric pressure chemical ionization (APCI) source.

## Isolation of $C_{16}$ compounds

A two liter culture of AM109 cells expressing *Sp*SodMT was extracted with cHex at room temperature. After filtration and evaporation of the solvent *in vacuo*, the organic extract (233.0 mg) was subjected to gravity column chromatography on silica gel, using cyclohexane (cHex) with increasing amounts of EtOAc as the mobile phase, to yield 7 fractions (1-7). Fraction 3 (24.0 mg) was purified by normal phase HPLC, using cHex/EtOAc (90:10) as eluent, to afford **1b** (0.4 mg). Fractions 4 (36.0 mg) and 5 (41.0 mg) were separately purified by normal phase HPLC, using cHex/EtOAc (80:20) and subsequently *n*-Hex/Me₂CO (82:18) as eluent, to afford **1a** (12.7 mg).

A two liter culture of yeast cells co-expressing *Sp*SodMT and Cyc2 in the presence of 10 mM methyl cyclodextrins was centrifuged and the medium was extracted with cHex at room temperature. After separation of the organic layer and evaporation of the solvent *in vacuo*, the organic extract was subjected to gravity column chromatography on silica gel, using cHex and subsequently $CH_2Cl_2$ as the mobile phase, to yield 4 fractions (1-4). Fraction 1 (3.4 mg) was purified by normal phase HPLC, using cHex/EtOAc (95:5) as eluent, to afford **1d** (2.1 mg). Fraction 2 (3.7 mg) was purified by normal phase HPLC, using cHex/EtOAc (90:10) as eluent, to afford **1b** (2.1 mg). Fraction 4 (3.6 mg) was purified by normal phase HPLC, using cHex/EtOAc (80:20) as eluent, to afford **1a** (2.3 mg).

A two liter culture of AM109 cells expressing *Sp*SodMT(Q57N) was extracted with cHex at room temperature. After filtration and evaporation of the solvent *in vacuo*, the organic extract (461.0 mg) was subjected to vacuum column chromatography on silica gel, using cHex with increasing amounts of EtOAc as the mobile phase, to yield 6 fractions (1-6). Fraction 4 (78.3 mg) was purified by normal phase HPLC, using *n*-Hex/EtOAc (80:20) as eluent, and subsequently by chiral HPLC, using *n*-Hex/i-PrOH (96:4) as eluent, to afford **1a** (1.2 mg), **2a** (3.0 mg) and **2c** (1.7 mg).

A two liter culture of AM109 cells expressing *Sp*SodMT(F58L) was extracted with cHex at room temperature. After filtration and evaporation of the solvent *in vacuo*, the organic extract (368.0 mg) was subjected to vacuum column chromatography on silica gel, using $CH_2Cl_2$ with increasing amounts of $Et_2O$ and finally 100% EtOAc as the mobile phase, to yield 6 fractions (1-6). Fractions 2 (8.7 mg) and 3 (12.6 mg) were separately purified by normal phase HPLC, using cHex/EtOAc (87:13) and cHex/Me₂CO (96:4) as eluent, and subsequently by chiral HPLC, using *n*-Hex/i-PrOH (98:2) as eluent, to afford **2c** (0.5 mg) and **3c** (0.7 mg).

A two liter culture of AM109 cells co-expressing *Sp*SodMT(Q57N) and Cyc2 was extracted with cHex at room temperature. After filtration and evaporation of the solvent *in vacuo*, the organic extract (300.0 mg) was subjected to reversed phase HPLC, using the following elution profile with a combination of solvent A ($H_2O$-MeCN 95:5 containing 0.1% formic acid) and solvent B ($H_2O$-MeCN 5:95 containing 0.1% formic acid): 0 min, 50% B; 30 min, 100% B; 50 min, 100% B, to afford a fraction containing compounds **1d** and **2d** that was subsequently purified with isocratic elution of 75% B to afford a mixture (1:1) of **1d** and **2d** (3.0 mg).

Three 1.5 L cultures of AM109 cells expressing *Sp*SodMT and *Ha*KS and PtAO were extracted with cHex at room temperature. After filtration and evaporation of the solvent *in vacuo*, the organic extract (200.5 mg) was subjected to vacuum column chromatography on silica gel, using cHex, with increasing amounts of EtOAc as the mobile phase, to yield 9 fractions (1–9), among which fraction 8 (50% EtOAc in cHex) was identified as compound **13OH-1b** (6.7 mg). Fractions 2 (5.1 mg) and 3 (124.9 mg) were combined and further purified by normal phase HPLC, using cHex/EtOAc (95:5) as eluent, and subsequently by chiral HPLC, using *n*-Hex/i-PrOH (99:1), to afford **1b** (70.0 mg) and **epi-1b** (1.7 mg). Fraction 4 (13.6 mg) was submitted to normal phase HPLC, using cHex/EtOAc (90:10) as eluent, to yield **1a** (9.5 mg) and **1c** (0.8 mg). Fraction 5 (18.3 mg) was subjected to normal phase HPLC, using cHex/EtOAc (80:20) as eluent, to afford **1a** (2.0 mg).

Six 1.5 L cultures of AM109 cells expressing *Sp*SodMT(F58L) and *Ha*KS and PtAO, as well as four 1.5 L cultures of AM109 cells expressing *Sp*SodMT(F58M) and *Ha*KS and PtAO were separately extracted with cHex at room temperature. After filtration and evaporation of the solvent *in vacuo*, the two organic extracts, displaying very similar chemical profiles, were pooled together (552.5 mg) and subjected to vacuum column chromatography on silica gel, using cHex with increasing amounts of EtOAc as the mobile phase, to yield six fractions (1-6). Fraction 2 was subjected to vacuum column chromatography on silica gel, using *n*-Hex with increasing amounts of EtOAc as the mobile

phase, to afford three fractions (2a-2c). Fraction 2c (70.5 mg) was subjected to normal phase HPLC using cHex/EtOAc (93:7) and subsequently chiral HPLC using *n*-Hex/i-PrOH (99:1) to afford **1b** (4.3 mg), **2b** (5.8 mg) and **3b** (10.1 mg). Fraction 3 (100.3 mg) was subjected to normal phase HPLC using cHex/EtOAc (85:15) and subsequently chiral HPLC using *n*-Hex/i-PrOH (99:1) to obtain **1b** (4.3 mg), **2b** (15.1 mg), **3b** (4.7 mg) and **4b** (16.2 mg). Fraction 4 (80.3 mg) was purified by normal phase HPLC using cHex/EtOAc (85:15) and subsequently cHex/Me$_2$CO (88:12) as eluent to afford compound **3c** (1.6 mg). Fraction 6 (39.9 mg) was separated by vacuum column chromatography on silica gel, using cHex with increasing amounts of EtOAc as the mobile phase, to afford six fractions (6a-6f). Fraction 6b (10.7 mg) was further purified by normal phase HPLC using *n*-Hex/EtOAc (50:50) as eluent to afford **9OH-3b** (2.3 mg). Fractions 6c (11.6 mg) and 6d (5.3 mg) were separately subjected to normal phase HPLC using *n*-Hex/EtOAc (40:60) as eluent to afford **5a** (0.6 mg), **5b** (1.0 mg) and ***epi*-5b** (0.8 mg).

Three 1.5 L cultures of AM109 cells expressing *Sp*SodMT(V273A) and *Ha*KS were extracted with cHex at room temperature. After filtration and evaporation of the solvent *in vacuo*, the organic extract (223.8 mg) was fractionated by vacuum column chromatography on silica gel, using cHex with increasing amounts of EtOAc as the mobile phase, to afford 9 fractions (1–9). Fraction 2 (81.6 mg) was subjected to normal phase HPLC, using cHex/EtOAc (95:5) as eluent, to yield **4b** (33.3 mg) and **9b** (1.8 mg). Fractions 3 (46.5 mg) and 4 (44.6 mg) were separately and repeatedly submitted to normal phase HPLC, using cHex/EtOAc (90:10) and cHex/Me$_2$CO (93:7) as eluent, to afford **1a** (9.5 mg), **1b** (2.0 mg), **4a** (18.9) and **4c** (0.8 mg). Fraction 7 (2.5 mg) was purified by normal phase HPLC, using cHex/EtOAc (80:20) as eluent, to afford **8b** (0.4 mg) and ***epi*-8b** (0.6 mg). Fraction 8 (6.6 mg) was subjected to normal phase HPLC, using cHex/EtOAc (60:40) as eluent, to yield **5a** (1.0 mg), **5b** (0.9 mg) and ***epi*-5b** (0.9 mg).

Four 1.5 L cultures of AM109 cells expressing *Sp*SodMT(F58M-V273A) and *Ha*KS, as well as four 1.5 L cultures of AM109 cells expressing *Sp*SodMT(F58M-L302S) and *Ha*KS were separately extracted with cHex at room temperature. After filtration and evaporation of the solvent *in vacuo*, the two organic extracts, displaying very similar chemical profiles, were pooled together (430.5 mg) and subjected to vacuum column chromatography on silica gel, using cHex with increasing amounts of EtOAc as the mobile phase, to yield six fractions (1–6). Fractions 2 and 3 were combined and fractionated with vacuum column chromatography on silica gel, using *n*-Hex with increasing amounts of EtOAc as the mobile phase, to afford four fractions (2a-2d). Fractions 2b (134.6 mg) and 2c (54.9 mg) were separately and repeatedly subjected to normal phase HPLC, using cHex/EtOAc (90:10) and subsequently *n*-Hex/EtOAc (90:10 and 92:8), and further purified by chiral HPLC, using *n*-Hex/i-PrOH (99:1), to afford **1b** (3.4 mg), **2b** (11.9 mg), **3b** (7.3 mg), **4b** (10.7 mg) and **7b** (6.7 mg). Fractions 4 (56.1 mg) and 5 (34.1 mg) were repeatedly subjected to normal phase HPLC, using cHex/EtOAc (85:15 and 80:20) and *n*-Hex/EtOAc (78:22), chiral HPLC, using *n*-Hex/i-PrOH (98:2 and 95:5), and reversed phase HPLC, using MeOH/H$_2$O (99:1), to afford **2a** (0.7 mg), **3a** (1.0 mg), **4a** (7.5 mg), **6a** (1.2 mg), **7a** (5.4 mg) and **7c** (0.8 mg). Fraction 6 (31.1 mg) was subjected to vacuum column chromatography on silica gel, using *n*-Hex with increasing amounts of EtOAc as the mobile phase, to obtain six fractions (6a-6f). Fractions 6b (3.1 mg), 6c (6.3 mg), 6d (4.5 mg) and 6e (1.1 mg) were separately subjected to normal phase HPLC, using cHex/EtOAc (50:50) and *n*-Hex/EtOAc (50:50) as eluent, to afford **5a** (1.6 mg), **5b** (2.3 mg), ***epi*-5b** (1.7 mg) and **10b** (0.8 mg).

### Structure elucidation of C$_{16}$ compounds

The chemical structures and relative configurations of the 28 C$_{16}$ compounds isolated in pure form (Supplementary Fig. 1) were determined on the basis of extensive analyses of their NMR and MS spectroscopic data (Supplementary Tables 9–14, Supplementary Figs. 16–230, Supplementary Note 1).

### Reporting summary

Further information on research design is available in the Nature Research Reporting Summary linked to this article.

## Data availability

Data supporting the findings of this work are available within the paper and its Supplementary Information files. A reporting summary for this Article is available as a Supplementary Information file. Previously reported structural data for pavine *N*-methyltransferase in complex with tetrahydropapaverine and *S*-adenosylhomocysteine, used as template in the modeling studies, is available at PDB under accession 5KOK [www.rcsb.org/structure/5KOK]. Previously reported structural data for farnesyl diphosphate (FPP), used in the docking studies, is available at ZINC [https://zinc.docking.org/substances/ZINC000012494625/]. Source data are provided with this paper.

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

## Acknowledgements

We would like to thank Dr. Fernando-Geu-Flores, Dr. Irini Pateraki and Dr. Simon Dusséaux (University of Copenhagen, Denmark) for critically reading the manuscript. We also thank Dr. David Ian Pattison (University of Copenhagen, Denmark) for support in metabolite analysis, Jack Olsen (University of Copenhagen, Denmark) for assisting in GC-MS-O and general metabolite analysis, and Kirsten Sjøstrøm (University of Copenhagen, Denmark) for support in GC-MS-O analysis. GC-MS-O data was generated using research infrastructure at the University of Copenhagen supported by FOODHAY (Food and Health Open Innovation Laboratory, Danish Roadmap for Research Infrastructure). This work was supported by the Novo Nordisk Foundation grants NNF16OC0021760 and NNF19OC0055204 (to SCK), NNF16OC0019554 (to CI), NNF18OC0031872 (to KM), and NNF16OC0021616 (to DS), and the Danish Council for Independent Research grants 7017-00275B and 0136-00410B (to SCK). Work at the National and Kapodistrian University of Athens was supported by the research projects BioNP (grant number 70/3/14685 to EI) and MARINOVA (grant number 70/3/14684 to VR).

## Author contributions

C.I.: conceived the initial part of the project, selected $Sp$SodMT as base of the study, identified TPSs and CYPs to synthesize the $C_{16}$ terpenoids, established production of $C_{16}$ terpenes in yeast, analyzed data, and drafted the manuscript. M.H.R.: constructed $Sp$SodMT mutants, carried out in vitro assays, assisted in cloning, produced and isolated compounds. A.K.: contributed to fractionation and isolation of compounds and NMR data analysis. Y.Z.: contributed to GC-MS-O analysis, CYP product analysis, compound isolation, NMR data analysis and structure elucidation. Y.-T.D.: contributed to compound isolation. M.H.: contributed to fractionation and isolation of compounds and NMR data analysis. K.M.: contributed to compound isolation. S.E.V.-L.: assisted in cloning, yeast cultivation, and product extraction. M.R.: performed molecular modeling of $Sp$SodMT and docking of substrate and co-substrate. P.G.: contributed to fractionation and isolation of compounds and NMR data analysis. D.S.: contributed to NMR data analysis and structure elucidation. M.A.P.: contributed to GC-MS-O and interpretation. W.L.P.B.: contributed to GC-MS-O analysis and training. V.R.: contributed to NMR data analysis and structure elucidation. E.I.: led the compound isolation and structural characterization efforts, analyzed the NMR data, drafted and revised the manuscript. S.C.K.: conceived and led the project, designed and coordinated research, designed the mutagenesis of $Sp$SodMT, analyzed the data, and wrote the manuscript.

## Competing interests

The authors declare no competing interests.
