## [Peer Review File · Nature Communications]

Expanding the terpene biosynthetic code with non-canonical 16 carbon atom building blocksReviewers' Comments:

Reviewer #1:

Remarks to the Author:

Ignia et al. explored the production of non-canonical 16-carbon building blocks leading to new-to-nature terpenes. The manuscript is divided into four main sections. The first section describes the production of pre-sodoriferyl diphosphate (PSPP) from the methyltransferase (SodMT) in yeast and its byproducts (1a – 1d) via acid hydrolysis or the action of native yeast enzymes. As stated in the text, SodMT has already been identified in the rhizobacterium *Serratia plymuthica* (Ref. 10). Moreover, it has been heterologously expressed in *E. coli* (Duell et al. *Microb. Cell Fact.* 2019;18(1):32. doi:10.1186/s12934-019-1080-6). Therefore, the first section, which describes the heterologous production of SodMT in *S. cerevisiae*, cannot be considered a novel approach.

The second section describes the production of new-to-nature C16 scaffolds via SodMT engineering. The authors constructed several SodMT variants producing different product profiles than the parental enzyme. Even though the authors mention that for some of the new compounds the production was efficient (e.g., 2b, line 197), no attempt was made to characterize their structure. Therefore, it is not clear whether their rational approach indeed resulted in seven distinct building blocks. Moreover, the conclusion that the obtained SodMT are dedicated synthases for each scaffold is not supported by the data. In vitro experiments of the SodMT variants would give a clearer view of their genuine products and relative correlation.

The third section describes the investigation of various terpene synthases that are able to convert the new C16 building blocks into new terpene scaffolds. The authors screened 14 terpene synthases, but only two (Cyc2 and HaKS) were able to accept the C16 building blocks. This is not surprising since both enzymes catalyze similar reactions to those observed in vivo (possibly by native yeast phosphatases). It should be noted that the Cyc2 products (1d – 6d) were also detected by the wild-type SodMT (or variants) in the yeast system (as shown in Figure 5), but the authors did not discuss this in the previous section. The promiscuous activity of Cyc2 and HaKS has been reported in the literature (e.g., Jia et al. *Metab. Eng.* 2016; 37:24-34. doi.org/10.1016/j.ymben.2016.04.001).

The fourth section describes the decoration of the new scaffolds by P450 oxidation. The authors investigated nine P450s, but only the promiscuous PtAO was found to act upon specific scaffolds. According to the authors, PtAO functionalized compounds 2b and 5b, which were not structurally characterized as mentioned above. Minor amounts of the new products were also found in control samples lacking expression of HaKS, eventually creating further confusion regarding the structure and origin of compounds 35 – 43, many of which appeared in trace amounts.

Overall, the manuscript cannot be accepted in its current form and further data are necessary to support the authors' conclusions.

Additional Comments:

1) In the Abstract and Introduction, there are contradictory sentences. On the one hand, the authors state "the chemical space occupied by specialized metabolites, and, thus, their application potential, is limited because their biosynthesis is based on only a handful of building blocks". On the other hand, "isoprenoid structures identified to date exceed 70,000 different compounds" and "terpene scaffolds to produce a vast diversity of compounds within each specific class."

Isoprenoids constitute one of the largest groups of natural products, and their chemical space is not limited. In fact, the enzyme used in this study (SodMT) has been derived from a natural source and is involved in the biosynthesis of the C16 terpene sodorifen. Decoration of terpene backbones with various functional groups (e.g., methylation) is well reported leading to limitless possibilities. Therefore, the authors should discuss the benefit of synthesizing new-to-nature terpene scaffolds with possible new applications instead of the "limited" complexity of this vast class of compounds.

2) Abstract: "Herein, we focus on isoprenoids and, by combining gene mining with protein and metabolic engineering".

There are no gene mining experiments in this study. If the authors have performed gene mining, they should provide the respective data.

3) Abstract: "construct yeast cells that selectively produce seven non-canonical isoprenoid building blocks", and Introduction (lines 73-74): "identify SpSodMT residues responsible for product-specificity and engineer enzyme variants that specifically synthesize six additional C16 prenyl diphosphate building blocks".

The experimental data do not support this conclusion. The multiple peaks in the chromatograms (Figure 4 and Suppl. Figure 12) clearly show no product specificity of the enzyme variants.

4) Introduction (lines 73-74): "In total, we produce over 60 new-to-nature isoprenoid compounds with C16 carbon skeletons".

Of the 65 compounds, only eight were structurally identified. The remaining 57 compounds are assumed to possess a C16 carbon framework based on their low-resolution MS spectra and postulations regarding the enzymes. Moreover, there are no high-resolution MS data for these compounds (except for 1a-1c). The lack of structural information makes the manuscript hard to follow and might eventually lead to the wrong conclusions. For example, in the 'Supplementary Note', the retention time of compound 38 is 13.44 min in L14 (extract?), whereas 13.27 min in U40. Undoubtedly, further structural data are necessary to support the conclusion that over 60 new-to-nature isoprenoid compounds with C16 carbon skeletons were produced.

5) Line 122: There is no Figure 2c in the text.

6) Figure 4b: The peak 2c and the peaks corresponding to the SPP scaffold are missing from the chromatogram.

7) Figure 5: In the chromatogram of SpSodMT wt, the olefin isomers (1d – 1f) were detected. However, this information was omitted from the discussion of Figure 4 and Suppl. Figure 12. The same applies to the olefin 2d, which is not shown in Figure 4b.

8) Line 237: The authors stated that "SpSodMT(F58V) became a dedicated PPP synthase, producing 77% of this building block". However, it should be noted that the intensity of the peaks is very low (near the baseline), suggesting that the production is too low, possibly due to decreased activity of the enzyme variant.

9) Line 244: The peaks of compounds 3g and 4c are not indicated in the SodMT (F58L) chromatogram (Figure 4d). The assignment of the peaks 3a-3c and 4a-4b was based on the structures of 3g and 4c and by comparison of their MS fragmentation pattern. This assignment is ambiguous considering the low-resolution MS spectrum and the similar structures of the produced carbon frameworks.

10) Line 250: The authors argued the formation of compounds 3c and 4c by analogy to the mechanism of 1c formation. However, compound 1c was not characterized, and its formation was based on an assumption (lines 124-127). Therefore, 1c cannot be used as an example for the characterization of compounds 3c and 4c.

11) Line 253: The oxidation of 3c by a yeast monooxygenase to produce 3g needs experimental confirmation.

12) Line 257: According to Supplementary Table 4, the fragment of m/z 109 refers to SPP but not to

compounds 5a and 5b.

13) Lines 274-277: The V273 variants are not highly specific for the synthesis of SPP. Multiple peaks are belonging to other scaffolds (1, 2, 6, and 7) in the chromatogram of the best variant V273A (Figure 4e).

14) Lines 285-290: In the Summary, the conclusion is not supported by the data. The authors have only characterized the scaffolds 1a and 2a, but not the scaffolds corresponding to 3a, 4a, 5a, 6a, and 7a. Therefore, the production of seven C16 building blocks remains an assumption. Again, it is clear from the chromatograms in Figure 4 that the SodMT variants are not dedicated synthases. This is especially true for the assumed scaffolds 6 and 7 (Figures 4g and f).

15) Lines 290-295: The authors support that for some of the variants, the total C16 production titer is higher than that of the wild-type enzyme. However, as mentioned above, the C16 production is divided into multiple peaks. Therefore, the individual titer for each of these peaks is very low, which is why most of them could not be characterized.

16) Line 372: The product 7d is not shown in the Suppl. Figure 27.

17) Lines 377 & 396: The peaks of the assumed Cyc2 and HaKS-derived products 8 – 31 are not shown in the corresponding chromatograms.

18) Lines 400-401: This conclusion is not supported by the data. The production of the 33 new terpenoids from Cyc2 and HaKS is an assumption.

19) Lines 421-422: PtAO was not efficient in oxidizing C16 scaffolds as minor amounts of the new products (32-43) were produced (clearly shown in Figure 6).

20) Lines 424-425: The levels of 1a were also depleted according to Figure 6.

21) Lines 426-427 & Figure 6: The product profiles of the SodMT variants with PtAO only should also be provided, as in Suppl. Figure 31.

22) Lines 468-479: The GC-MS-olfactometry section needs further clarification. Only compounds 1b and 3b were found to possess odor properties from the whole extract?

23) Suppl. Table 1: High-resolution MS spectra of compounds 1a-1d, 2a-2d, 3g, and 4g should be provided.

24) Suppl. Table 4: The number of several products is not indicated.

25) For the characterized compounds 1a, 1b, 1d, 2a, 2c, 2d, 3g, and 4c ¹³C, COSY, and HMBC spectra should be provided as well. This is especially important for compound 3g, which gave a poor ¹H NMR spectrum.

26) HR-MS spectra of the new compounds 1a, 1b, 1d, 2a, 2c, 2d, 3g, and 4c should be provided to support the structures.

27) Suppl. Figure 5: From the ¹H NMR, it seems that 1b is a mixture of two compounds.

28) Suppl. Figure 7: No reduction of 1a is observed to support the hypothesis of production of compound 1c from 1a.

29) In Suppl. Figure 10, the amount of 1b is very low compared to that of Figure 2. The authors

should explain this discrepancy between results.

30) Supplementary Figure 12: In the chromatogram of SpSodMT (L302S), the peak with a retention time of 15min was not identified.

31) Suppl Info: The sequence of the genes corresponding to the investigated enzymes should be provided.

Reviewer #2:

Remarks to the Author:

The authors have submitted a manuscript reporting the production of isoprenoids by combining gene mining with protein and metabolic engineering. Yeast cells were selected to selectively produce seven non-canonical isoprenoid building blocks with 16 carbon atoms, the result of a SAM-dependent methyltransferase, and these metabolic pathways were combined with terpene synthases to produce terpenoids with C16 backbones that were further functionalized using a cytochrome P450-containing oxidase. In addition, olfactory properties were collected.

This concept is based on purely biosynthetic/biotechnological steps and complements very well hybrid strategies based on chemically prepared prenyl diphosphate derivatives transformed and further modified by terpene synthases, either enzymatically or chemically. The authors completely omit this very closely related area of research and Allemann, Dickschat, Kirschning, Cane and other groups are not mentioned at all. It would have been easy to do this because, first, there are various recent papers on non-natural terpenoids in this area as well, also with interesting olfactory properties, and second, a review on this topic was recently published in *Nat. Prod. Res.* 2020, 37, 1080 - 1097.

Nevertheless, I consider the present work highly interesting from a conceptual point of view, and in particular the mutagenesis studies performed are quite successful.

However, I have to raise some critical points:

1. the authors emphasize that they have prepared more than 60 new products, but in fact it seems that only a few have been isolated and structurally confirmed. Structural elucidation must also include stereochemical aspects. However, many of these molecules are only postulated, e.g., by GC-MS. This is not the usual scientific claim in natural product chemistry and the authors should limit themselves to real examples, since their research program should generate new chemical entities. The authors either need to delete many examples or take more care in isolation and structure elucidation. GC-MS analyses alone are insufficient. A very suitable method to separate mixtures of terpenes of this molecular weight is preparative GC.

2. The products of SpSodMT terpene synthase and subsequent terpene synthases are very simple products, mainly cleavage of diphosphate and trapping of water, which are basically hydrolysis products. For these, it must be shown that they are enzymatically generated products or are the result of chemical hydrolysis. Were the tests performed with or without negative controls?

3. The products formed by the terpene synthases Cyc2 and HaKS are also already present in small amounts in the SpSodMT -> here the question arises even more strongly whether they are really biotransformation products or whether they are simply formed by hydrolysis. For HaKS, it is stated that the same products are also present in the negative control, but in lower amounts -> were the negative controls really incubated for the same amount of time?

4. Why are the same products formed in Cyc2 and HaKS, when it was originally noted that the two TCs were selected because they rely on different mechanisms?

5. Were the mutation sites selected based on a homology model or crystal structure, this is not

clarified in the manuscript.

6 -The authors did not isolate any product formed during P450 oxidation, only conjectures are made about the products formed. One can delete the section de facto.

7. The bibliography is completely inadequate, as already mentioned above.

8. The introduction about the role and abundance of terpenes is rather general and the text part up to the description of the actual experiments should be shortened considerably. On the other hand, the necessity of diversity approaches is discussed without comparatively mentioning existing alternatives.

9. The selection of terpene synthases was limited to those forming five- or six-membered ring systems. Chemically, synthases that generate more complex oligocyclic products are of greater interest because, on the one hand, they are often more easily accessible biotechnologically than by chemical means and, on the other hand, they have a broader biological activity profile. Overall, this is conceptually a very interesting paper. However, the quality in terms of handling and analysis of the products has been unsatisfactorily executed. The authors should devote considerable greater care to the chemical aspects of their work, since the motivation for this research is solely to produce new terpenoids that would not otherwise be readily available, especially with respect to the handling of unnatural diphosphate-containing substrates.

The manuscript in its present form is not suitable for publication in Nature Communications.

Reviewer #3:

Remarks to the Author:

The manuscript by Ignea et al. entitled "Expanding the terpene biosynthetic code with non-canonical 16 carbon atom building blocks" is a well written description of a study in which the authors engineer yeast to produce non-canonical sesquiterpene derivatives in an effort to expand the availability of natural-product like organic compounds. The study follows nicely work from the same group in which a similar approach was used to generate methylated derivatives of monoterpenes. They begin by engineering yeast to express wild-type and mutant forms of a methyltransferase from the rhizobacterium *Serratia plymuthica*, SpSodMT, that produce novel cyclic, methylated sesquiterpene derivatives as substrates for two clerodane type diterpenoid synthases, terpentetriene synthase (Cyc2) from *Kitasatospora griseola*, and kolavelool synthase (HaKS) from *Herpetosiphon aurantiacus*. Finally, they complete the study by introducing a P-450, abietadiene oxidase (CYP720B1) from *Pinus taeda*, for oxidative adornment of the modified sesquiterpene scaffolds. All told they generate more than 60 new compounds that the authors coin "novoterpenes". The study is well-designed and executed. The results represent a clear advance in the field. The work is timely and will appeal to the broad readership of Nature Communications.

Response to reviewers' comments (NCOMMS-20-51374)

We would like to sincerely thank the reviewers for their thorough and constructive comments. We have now addressed all major concerns with additional experimental evidence.

1. We have performed considerable work on compound isolation and structure elucidation and now provide comprehensive NMR data for 26 compounds to confirm the synthesis of eight new non-canonical C₁₆ isoprenoid building blocks, in addition to PSPP.
2. We have confirmed with *in vitro* assays that KS and Cyc2 utilize the new building blocks as substrates to produce C₁₆ terpenoids (and also explain the background production seen during yeast cultivation in acidic media).
3. We have isolated and elucidated the structures of two products of P450-mediated oxidation of C₁₆ skeletons.
4. We used *in vitro* assays to analyze the product profile of the *SpSodMT* variants and confirm that this correlates well with the C₁₆ compound profile produced by yeast cells expressing the specific mutants.
5. We have clarified points of misunderstanding, provided additional bibliographic reference where need, and revised/refined other claims taking into consideration the reviewers' concerns.

The extensive additional structural analysis efforts have greatly strengthen the research presented and the reviewers' remarks and suggestions have considerably improved the quality of the manuscript. Overall, we feel that we now provide convincing evidence to substantiate the main claims of the manuscript and satisfy all other concerns the reviewers have raised. Please find below our point-by-point response.

Reviewer #1:

Major comments:

Comment 1.1: Ignea et al. explored the production of non-canonical 16-carbon building blocks leading to new-to-nature terpenes. The manuscript is divided into four main sections. The first section describes the production of pre-sodorifenyl diphosphate (PSPP) from the methyltransferase (SodMT) in yeast and its byproducts (1a – 1d) via acid hydrolysis or the action of native yeast enzymes. As stated in the text, SodMT has already been identified in the rhizobacterium *Serratia plymuthica* (Ref. 10). Moreover, it has been heterologously expressed in *E. coli* (Duell et al. *Microb. Cell Fact.* 2019;18(1):32. doi:10.1186/s12934-019-1080-6). Therefore, the first section, which describes the heterologous production of SodMT in *S. cerevisiae*, cannot be considered a novel approach.

Response: The first section aims to describe the yeast system on which the remaining of the study is based. Although *SpSodMT* has indeed been expressed in *E. coli*, its functional expression in yeast had not been

confirmed before, neither had the establishment of a complete heterologous biosynthetic pathway converting sugar to *Sp*SodMT-derived products. Furthermore, in this section, we confirm that this heterologous C₁₆ pathway is orthogonal to the yeast metabolism, which is an important element of this work and an essential step in order to achieve eventual bioproduction of the desired novoterpenes. Moreover, in this part, we also explain how we used hydrolysis of the synthesized C₁₆ prenyl diphosphates by the acidified yeast cultivation medium, or by yeast phosphatases, as a tool to detect their synthesis. Therefore, this section is essential for the reader to follow the approach that was used in the rest of the manuscript.

In response to this comment, we have now edited this section to be as short and concise as possible without losing essential information (**lines 100-118**; line numbers in this document correspond to the version of the revised manuscript with changes highlighted). We also simplified the main figure describing the results of this section, **Fig. 2**, by removing panel b. Furthermore, a citation to the work of Duell and colleagues was added **in line 60**.

Comment 1.2: The second section describes the production of new-to-nature C₁₆ scaffolds via SodMT engineering. The authors constructed several SodMT variants producing different product profiles than the parental enzyme. Even though the authors mention that for some of the new compounds the production was efficient (e.g., 2b, line 197), no attempt was made to characterize their structure. Therefore, it is not clear whether their rational approach indeed resulted in seven distinct building blocks.

Response: In the revised manuscript, we were able to optimize our methodology and overcome previous challenges relating to the isolation of the target compounds. Thus, we now report the chemical structures of a total of **26** C₁₆ compounds. Based on this extensive structural analysis, we deduce the structure of eight new C₁₆ prenyl diphosphate building blocks. We confirm the synthesis of different products of enzymatic conversion of these prenyl diphosphates in our engineered cells or in in vitro enzyme reactions. These results have now been included in the revised results section (**lines 108-115, 187-193, 230-234, 240-250, 258-276, 412-427**), and the materials and methods section (**lines 539-540, 746-872**). The figures and the Supplementary information have also been modified accordingly to accommodate the new findings. A **Supplementary Note** with all the NMR, LR-EI-MS and HR-APCI-MS data of the isolated compounds, as well as information relating to the structure elucidation is also included in the revised version of the manuscript.

Comment 1.3: Moreover, the conclusion that the obtained SodMT are dedicated synthases for each scaffold is not supported by the data. In vitro experiments of the SodMT variants would give a clearer view of their genuine products and relative correlation.

Response: To confirm that the product profiles obtained in the *in vivo* experiments indeed reflect the product specificity of the enzyme variants, we carried out *in vitro* experiments with different SpSodMT variants and FPP and SAM as co-substrates. In **Supplementary Figure 9 and Supplementary Table 7**, we show that the *in vitro* product specificities of the mutants correspond closely to the product profile obtained from yeast cells in acidic media. Thus, for example, SpSodMT(V273A), is indeed an enzyme with more than 70% selectivity for weylantdenyl diphosphate (WPP), while SpSodMT(F58M-L302Q) indeed produces kimlarsenyl diphosphate (KPP) as the predominant product and SpSodMT(F58L) has strong preference for TPP. However, SpSodMT(F58V) exhibited poor activity (or instability) and we could not obtain convincing *in vitro* data to confirm its apparent preference for PPP synthesis in yeast. We present these findings in **lines 271-276** of the revised manuscript.

Comment 1.4: The third section describes the investigation of various terpene synthases that are able to convert the new C16 building blocks into new terpene scaffolds. The authors screened 14 terpene synthases, but only two (Cyc2 and HaKS) were able to accept the C16 building blocks. This is not surprising since both enzymes catalyze similar reactions to those observed in vivo (possibly by native yeast phosphatases). It should be noted that the Cyc2 products (1d – 6d) were also detected by the wild-type SodMT (or variants) in the yeast system (as shown in Figure 5), but the authors did not discuss this in the previous section. The promiscuous activity of Cyc2 and HaKS has been reported in the literature (e.g., Jia et al. *Metab. Eng.* 2016; 37:24-34. doi.org/10.1016/j.ymben.2016.04.001).

Response: The high background observed in the experiments with Cyc2 and HaKS was due to the acidic pH of the yeast media. As explained in **lines 105-109 of the manuscript**, in these experiments, we used non-buffered media to grow the yeast cultures because, under these conditions, it is facile to observe synthesis of prenyl diphosphates by determining the presence of acid hydrolysis products of the prenyl diphosphate in the medium. For consistency, in the results presented subsequently in the initial manuscript, the HaKS and Cyc2 experiments were carried out under the same (acidic) conditions, and this resulted in the observation of considerable background production of the same compounds. In the revised version, we repeated the experiments using buffered media, which eliminated the background and convincingly showed that synthesis of the Cyc2 and HaKS products is specific to the presence of these enzymes (shown now in **revised Fig. 5**). To further confirm the specific synthesis of these compounds, we also carried out *in vitro* experiments with SpSodMT variants and HaKS or Cyc2 using FPP and SAM as co-substrates,

where we confirm that the specific products are only synthesized when the terpene synthase is present. These results are now included as **Supplementary Figures 13-14**.

Indeed, the promiscuous activity of Cyc2 and HaKS was known, but it was regarding C₂₀ substrates. This was one of the reasons these two enzymes were among those selected for this study. The text was adjusted to make this more obvious and the proposed citation was also added in **line 318**.

Comment 1.5: The fourth section describes the decoration of the new scaffolds by P450 oxidation. The authors investigated nine P450s, but only the promiscuous PtAO was found to act upon specific scaffolds. According to the authors, PtAO functionalized compounds 2b and 5b, which were not structurally characterized as mentioned above. Minor amounts of the new products were also found in control samples lacking expression of HaKS, eventually creating further confusion regarding the structure and origin of compounds 35 – 43, many of which appeared in trace amounts.

Response: We have now isolated two PtAO-functionalized compounds and characterized them as **13-hydroxy-presodorifelool** and **9-hydroxy-thorvaldselool**, confirming that PtAO can indeed accept C₁₆ terpenoids to yield further decorated products. The isolation and characterization of these compounds have now been included in the **results (lines x-y)**, **methods (lines k-l)**, and **in the Supplementary Note**. The section related to PtAO-mediated oxidation of non-canonical C₁₆ terpenoids has now been streamlined to focus on the isolated compounds.

Regarding the minor amounts found in samples without HaKS, these are due to the production of the tertiary alcohol substrates in acidic media, as explained in response to comment 1.5 and also in the text of the first submission (**lines 424-426**).

Additional Comments:

Comment 1.6: In the Abstract and Introduction, there are contradictory sentences. On the one hand, the authors state “the chemical space occupied by specialized metabolites, and, thus, their application potential, is limited because their biosynthesis is based on only a handful of building blocks”. On the other hand, “isoprenoid structures identified to date exceed 70,000 different compounds” and “terpene scaffolds to produce a vast diversity of compounds within each specific class.” Isoprenoids constitute one of the largest groups of natural products, and their chemical space is not limited. In fact, the enzyme used in this study (SodMT) has been derived from a natural source and is involved in the biosynthesis of the C₁₆ terpene sodorifen. Decoration of terpene backbones with various functional groups (e.g., methylation) is well reported leading to limitless possibilities. Therefore, the

authors should discuss the benefit of synthesizing new-to-nature terpene scaffolds with possible new applications instead of the “limited” complexity of this vast class of compounds.

Response: We have followed the reviewer’s advice and re-written this part of the introduction to simplify and streamline the message, focusing on expanding the structural complexity of terpenoids. The revised introduction now covers **lines 1-28** of the manuscript.

Comment 1.7: Abstract: ”Herein, we focus on isoprenoids and, by combining gene mining with protein and metabolic engineering”. There are no gene mining experiments in this study. If the authors have performed gene mining, they should provide the respective data.

Response: This expression has now been removed from the revised abstract.

Comment 1.8: Abstract: ”construct yeast cells that selectively produce seven non-canonical isoprenoid building blocks”, and Introduction (lines 73-74): “identify SpSodMT residues responsible for product-specificity and engineer enzyme variants that specifically synthesize six additional C16 prenyl diphosphate building blocks”. The experimental data do not support this conclusion. The multiple peaks in the chromatograms (Figure 4 and Suppl. Figure 12) clearly show no product specificity of the enzyme variants.

Response: It is true that although some variants are quite promiscuous synthases, we have been able to develop and confirm by *in vitro* experiments that several engineered synthases have considerable selectivity for the synthesis of specific building blocks. We have now rephrased the text to be more precise regarding this claim and to clarify that it is selectivity, and not absolute specificity, that we are referring to. This has now been changed in the abstract and in the introduction **in lines 65-66**

Comment 1.9: Introduction (lines 73-74): “In total, we produce over 60 new-to-nature isoprenoid compounds with C16 carbon skeletons”. Of the 65 compounds, only eight were structurally identified. The remaining 57 compounds are assumed to possess a C16 carbon framework based on their low-resolution MS spectra and postulations regarding the enzymes. Moreover, there are no high-resolution MS data for these compounds (except for 1a-1c). The lack of structural information makes the manuscript hard to follow and might eventually lead to the wrong conclusions. For example, in the ‘Supplementary Note’, the retention time of compound 38 is 13.44 min in L14 (extract?), whereas 13.27 min in U40. Undoubtedly, further structural data are necessary to support the conclusion that over 60 new-to-nature isoprenoid compounds with C16 carbon skeletons were produced.

Response: We expanded the structural characterization effort to now provide NMR data for a total of 26 different C₁₆ compounds and we accompany them with high-resolution MS data. All structural characterization data are now provided as a separate **Supplementary Note**.

We have revised the claim **in the abstract** and **in lines 70-72** of the introduction to refer to the synthesis of only the 26 compounds that we have fully characterized and to clarify that we do not have structural information for the remaining products whose isolation has not been possible at this stage.

Comment 1.10: Line 122: There is no Figure 2c in the text.

Response: Corrected. Thank you.

Comment 1.11: Figure 4b: The peak 2c and the peaks corresponding to the SPP scaffold are missing from the chromatogram.

Response: Figure 4b has now been revised to include annotation of all relevant peaks.

Comment 1.12: Figure 5: In the chromatogram of SpSodMT wt, the olefin isomers (1d – 1f) were detected. However, this information was omitted from the discussion of Figure 4 and Suppl. Figure 12. The same applies to the olefin 2d, which is not shown in Figure 4b.

Response: The olefin isomers in the strains lacking a terpene synthase originate from the hydrolysis of the substrate and they are produced at considerably lower levels than when Cyc2 is present. Thus, they were initially not shown/discussed in Fig. 4. We have now indicated some of the olefin isomers in the **revised Fig. 4** (in the cases where their presence can readily be observed; for example 2d in panel b) but we still consider that it is more relevant to discuss their synthesis in the next section and in conjunction with revised **Figure 5** and **Supplementary Figures 11-12** (revised manuscript, **lines 259-278**).

Comment 1.13: Line 237: The authors stated that “SpSodMT(F58V) became a dedicated PPP synthase, producing 77% of this building block”. However, it should be noted that the intensity of the peaks is very low (near the baseline), suggesting that the production is too low, possibly due to decreased activity of the enzyme variant.

Response: Indeed, this enzyme appears to be weak or less stable. We have made this comment now in the text, in **lines 223, 271** and **274-276** of the revised version.

Comment 1.14: Line 244: The peaks of compounds 3g and 4c are not indicated in the SodMT (F58L) chromatogram (Figure 4d). The assignment of the peaks 3a-3c and 4a-4b was based on the structures of 3g and 4c and by comparison of their MS fragmentation pattern. This assignment is ambiguous considering the low-resolution MS spectrum and the similar structures of the produced carbon frameworks.

Response: With the additional structures reported in the revised version, all eight new building blocks are now supported with relevant structures. The figures have been revised accordingly and this comment has now been resolved by providing additional structural data.

Comment 1.15: Line 250: The authors argued the formation of compounds 3c and 4c by analogy to the mechanism of 1c formation. However, compound 1c was not characterized, and its formation was based on an assumption (lines 124-127). Therefore, 1c cannot be used as an example for the characterization of compounds 3c and 4c.

Response: We have addressed this comment with additional compound isolation and characterization and now provide NMR data for three different type c compounds (1c, 2c, and 3c) (see also response to comment 1.14).

Comment 1.16: Line 253: The oxidation of 3c by a yeast monooxygenase to produce 3g needs experimental confirmation.

Response: Because the additional structural data obtained during the revision provided adequate support for the characterization of the new building blocks, the isolation and structural characterization of previous compound 3g is not included in the revised version.

Comment 1.17: Line 257: According to Supplementary Table 4, the fragment of m/z 109 refers to SPP but not to compounds 5a and 5b.

Response: **Supplementary Table 3** and **Supplementary Note** have now been updated with the new compounds and corresponding spectra.

Comment 1.18: Lines 274-277: The V273 variants are not highly specific for the synthesis of SPP. Multiple peaks are belonging to other scaffolds (1, 2, 6, and 7) in the chromatogram of the best variant V273A (Figure 4e).

Response: Although *SpSodMT(V273A)* makes more than one building blocks, as confirmed both by *in vitro* and *in vivo* studies, WPP accounts to more than 75% of its products. We used the term specific in its broader sense to describe the strong preference for the synthesis of one of the products. In response to this comment, we have now rephrased the text where possible to use alternative expressions to denote this preference. (e.g. **in lines 266-274** of the revised manuscript).

Comment 1.19: Lines 285-290: In the Summary, the conclusion is not supported by the data. The authors have only characterized the scaffolds 1a and 2a, but not the scaffolds corresponding to 3a, 4a, 5a, 6a, and 7a. Therefore, the production of seven C16 building blocks remains an assumption. Again, it is clear from the chromatograms in Figure 4 that the SodMT variants are not dedicated synthases. This is especially true for the assumed scaffolds 6 and 7 (Figures 4g and f).

Response: We now provide new structural data that confirm the synthesis of eight new prenyl diphosphate substrates. Regarding the question of specificity, please see response to comment 1.18, above, and to comment 1.3 for the *in vitro* studies carried out. This section (**lines 266-276**) has now been revised to include the new structural information and clarify the issue of product preference/selectivity.

Comment 1.20: Lines 290-295: The authors support that for some of the variants, the total C16 production titer is higher than that of the wild-type enzyme. However, as mentioned above, the C16 production is divided into multiple peaks. Therefore, the individual titer for each of these peaks is very low, which is why most of them could not be characterized.

Response: Initially, the main problem in the isolation and characterization of more products were caused by the poor stability and high volatility of many compounds. We have since found the technical solutions that enabled the isolation of considerably more compounds to support the main claims of this work. Using the isolated compounds as standards (**lines 595-608**), we have now recalculated the production titers and confirm that several engineered synthases are indeed quite efficient, in comparison to the wild-type *SpSodMT* enzyme, in the production of C₁₆ precursors in yeast. The product titers reported in the text and the corresponding tables and figures in the Supplementary Information have been updated with the new values (**Supplementary Tables 9, 10, 11; Supplementary Figures 7 & 10**).

Comment 1.21: Line 372: The product 7d is not shown in the Suppl. Figure 27.

Response: Because of the additional structural information obtained during manuscript revision, this figure is no longer part of the Supplementary Information.

Comment 1.22: Lines 377 & 396: The peaks of the assumed Cyc2 and HaKS-derived products 8 – 31 are not shown in the corresponding chromatograms.

Response: Because of the new structural information obtained in the revision phase many of the compounds in the Cyc2 and HaKS chromatograms have now been characterized. We are providing new figures in the revised manuscript and Supplementary Information. All the Cyc2 and HaKS produced compounds are now indicated in **revised Fig. 5, Supplementary Fig 11; Supplementary Fig. 12.**

Comment 1.23: Lines 400-401: This conclusion is not supported by the data. The production of the 33 new terpenoids from Cyc2 and HaKS is an assumption.

Response: In light of the new structural data, this statement was rewritten to clarify which of the Cyc2 and HaKS products were characterized and which were tentatively assigned (**lines 387-388**).

Comment 1.24: Lines 421-422: PtAO was not efficient in oxidizing C16 scaffolds as minor amounts of the new products (32-43) were produced (clearly shown in Figure 6).

Response: Considering the general efficiency of different plant P450s tested in the specific yeast system in our lab, PtAO's activity was indeed quite efficient. To avoid this quantitative statement, this sentence was rephrased in the revised manuscript (**lines 410-411**).

Comment 1.25: Lines 424-425: The levels of 1a were also depleted according to Figure 6.

Response: This is correct. However, this comment has been removed in the revised text, which now focuses on the characterization of the isolated PtAO products.

Comment 1.26: Lines 426-427 & Figure 6: The product profiles of the SodMT variants with PtAO only should also be provided, as in Suppl. Figure 31.

Response: The product profiles of the SpSodMT variants with PtAO only are now included in the revised **Supplementary Figure 15.**

Comment 1.27: Lines 468-479: The GC-MS-olfactometry section needs further clarification. Only compounds 1b and 3b were found to possess odor properties from the whole extract?

Response: Compounds **1b** and **4b** (new numbering) were the ones with odor properties among the C₁₆ products of the extracts. The analysis also revealed the presence of canonical terpenoids with known odorant properties in the mix, such as nerolidol or linalool, or the characteristic yeast metabolite phenylethyl alcohol, confirming the validity of the approach. The manuscript text was amended accordingly (**lines 469-472**).

Comment 1.28: Suppl. Table 1: High-resolution MS spectra of compounds 1a-1d, 2a-2d, 3g, and 4g should be provided.

Response: We now provide high-resolution MS data for all the isolated compounds in the **Supplementary Note**.

Comment 1.29: Suppl. Table 4: The number of several products is not indicated.

Response: This table has now been revised accordingly.

Comment 1.30: For the characterized compounds 1a, 1b, 1d, 2a, 2c, 2d, 3g, and 4c ¹³C, COSY, and HMBC spectra should be provided as well. This is especially important for compound 3g, which gave a poor ¹H NMR spectrum.

Response: We now provide extensive data, including 1D and 2D NMR spectra, as well as NMR data in tabular form and all relevant information (COSY and HMBC correlations, as well as NOE cross-peaks) pertaining to the structural characterization of the different isolated products, in the **Supplementary Note**.

Comment 1.31: HR-MS spectra of the new compounds 1a, 1b, 1d, 2a, 2c, 2d, 3g, and 4c should be provided to support the structures.

Response: Same as comment 1.28.

Comment 1.32: Suppl. Figure 5: From the ¹H NMR, it seems that 1b is a mixture of two compounds.

Response: Indeed, in the originally submitted supplementary information, **1b** was a mixture of epimers at C-3. In the revised version it can be seen that both epimers (**1b** and *epi-1b*) have been isolated in pure form. Please see detailed characterization of **1b** and *epi-1b* in the **Supplementary Note**.

Comment 1.33: Suppl. Figure 7: No reduction of **1a** is observed to support the hypothesis of production of compound **1c** from **1a**.

Response: The structure of **1c** has now been confirmed by NMR and agrees with the proposed mechanism. In the specific figure, overexpression of Oye2p, which is already present in yeast cells, results in the doubling of the amount of **1c**. Reduction of **1a** may not have been observed in the specific sample because these experiments are carried out using yeast cells and the **1a** levels may fluctuate between samples because, for example, culture variations that can affect the activity of endogenous phosphatases. For this reason, what is more relevant is the ratio between **1c** and **1a**. To confirm that this was the case, we re-examined the complete dataset and confirmed that in other biological replicates the decrease in **1a** was readily observed. We have now updated **Supplementary Fig. 1** with the new data.

Comment 1.34: In Suppl. Figure 10, the amount of **1b** is very low compared to that of Figure 2. The authors should explain this discrepancy between results.

Response: In Supplementary Figure 10, the cells were subjected to non-saponifiable lipid extraction to separate and characterize the sterol fraction. The media were removed and only the cells were harvested. Thus, the low levels of **1b** present are expected because **1b** is found mostly in the culture medium as a product of acid hydrolysis of PSPP. On the contrary, **1a** is produced by the action of yeast phosphatases, such as Lpp1p or Dpp1p, and, because of this, it is expected to primarily be found inside the cells or associated with the cell membrane. This explanation is now included in the legend of **Supplementary figure 3**.

Comment 1.35: Supplementary Figure 12: In the chromatogram of SpSodMT (L302S), the peak with a retention time of 15min was not identified.

Response: This peak appears to be an indole-containing compound from yeast. The legend of **Supplementary Figure 5** was adjusted to explain this.

Comment 1.36: Suppl Info: The sequence of the genes corresponding to the investigated enzymes should be provided.

Response: We provide the corresponding sequence accession numbers for the all investigated enzymes in **Supplementary Table 8.**

Reviewer #2:

Comment 2.1: The authors have submitted a manuscript reporting the production of isoprenoids by combining gene mining with protein and metabolic engineering. Yeast cells were selected to selectively produce seven non-canonical isoprenoid building blocks with 16 carbon atoms, the result of a SAM-dependent methyltransferase, and these metabolic pathways were combined with terpene synthases to produce terpenoids with C16 backbones that were further functionalized using a cytochrome P450-containing oxidase. In addition, olfactory properties were collected.

This concept is based on purely biosynthetic/biotechnological steps and complements very well hybrid strategies based on chemically prepared prenyl diphosphate derivatives transformed and further modified by terpene synthases, either enzymatically or chemically. The authors completely omit this very closely related area of research and Allemann, Dickschat, Kirschning, Cane and other groups are not mentioned at all. It would have been easy to do this because, first, there are various recent papers on non-natural terpenoids in this area as well, also with interesting olfactory properties, and second, a review on this topic was recently published in Nat. Prod. Res. 2020, 37, 1080 - 1097.

Response: It was not been our intention to omit this work. We had restricted our introduction to synthetic biology approaches and whole-cell isoprenoid-producing systems. We have now revised the introduction to incorporate many of the proposed chemo-enzymatic studies with alternative substrates, which indeed add further support to our findings (**lines 34-37** of the revised manuscript).

Comment 2.2: Nevertheless, I consider the present work highly interesting from a conceptual point of view, and in particular the mutagenesis studies performed are quite successful.

Response: Thank you!

However, I have to raise some critical points:

Comment 2.3: the authors emphasize that they have prepared more than 60 new products, but in fact it seems that only a few have been isolated and structurally confirmed. Structural elucidation must also include stereochemical aspects. However, many of these molecules are only postulated, e.g., by GC-MS. This is not the usual scientific claim in natural product chemistry and the authors should limit themselves to real examples, since their research program should generate new chemical entities. The authors either need to delete many examples or take more care in isolation and structure elucidation. GC-MS analyses alone are insufficient. A very suitable method to separate mixtures of terpenes of this molecular weight is preparative GC.

Response: As already explained in response to the comments by reviewer #1, this issue has been addressed with additional compound isolation and characterization. In the revised manuscript, we provide substantial structural evidence for 26 C₁₆ terpenoid compounds (**Figures 1 and 3; Supplementary Tables 2&3, Supplementary Note**). We have also rephrased the manuscript to clearly explain which compounds have been structurally characterized and which have been assigned tentatively in the **abstract**, introduction (**lines 70-72**), and discussion (**line 513**).

Comment 2.4: The products of SpSodMT terpene synthase and subsequent terpene synthases are very simple products, mainly cleavage of diphosphate and trapping of water, which are basically hydrolysis products. For these, it must be shown that they are enzymatically generated products or are the result of chemical hydrolysis. Were the tests performed with or without negative controls?

Response: Yes. The experiments with terpene synthases were carried out including negative controls. The main reason for the high background is the hydrolysis of the building blocks under acidic conditions, as described in the text (**lines 105-109, 424-426**) and in the response to comment 1.5 (reviewer #1). We have addressed this issue by repeating the experiments using buffered media, and the reduction in the background is evident in the **revised Figure 5**.

Furthermore, we performed *in vitro* experiments to show that when the SpSodMT variants are combined with HaKS or Cyc2, the combination of the two enzymes can lead to synthesis of the specific C₁₆ products from FPP. These experiments have now been included in **lines 376-378 and 383-385** of the text and in **Supplementary Figures 13-14**.

Comment 2.5: The products formed by the terpene synthases Cyc2 and HaKS are also already present in small amounts in the SpSodMT -> here the question arises even more strongly whether they are really biotransformation products or whether they are simply formed by hydrolysis. For HaKS, it is stated that the same products are also present in the negative control, but in lower amounts -> were the negative controls really incubated for the same amount of time?

Response: Yes. The incubation time was identical in these cases. The explanation here is the acidic pH of the medium (see response to comment 2.4 above and also the response to reviewer #1). As explained above, we now provide experiments in buffered media and *in vitro* enzyme assays to confirm that Cyc2 and HaKS are indeed involved in this transformation.

Comment 2.6: Why are the same products formed in Cyc2 and HaKS, when it was originally noted that the two TCs were selected because they rely on different mechanisms?

Response: We observed that Cyc2 could also produce low amounts of the tertiary alcohol products of the C₁₆ diphosphates (**Revised Figure 5**). This activity of Cyc2 had not been reported before. This has now been clarified in the revised text, **lines 324-332, 376-378, and Supplementary Figure 13**.

Comment 2.7: Were the mutation sites selected based on a homology model or crystal structure, this is not clarified in the manuscript.

Response: Yes, indeed, the mutants were selected based on a homology model. Details about the selection of template and the extensive methodology followed to construct the model were included in the results and methods sections of the original submission (now **lines 159-161** (Results) and **723-738** (Methods) of the revised version).

Comment 2.8: The authors did not isolate any product formed during P450 oxidation, only conjectures are made about the products formed. One can delete the section de facto.

Response: Now we have isolated two of the P450 products (13-hydroxy-presodorifelool and 9-hydroxy-thorvaldselool) and have revised this section accordingly to focus on the isolated compounds (**lines 410-437**).

Comment 2.9: bibliography is completely inadequate, as already mentioned above.

Response: As discussed above, we had initially focused on the biotechnological angle of this work. New bibliography has now been added to also discuss chemoenzymatic studies (**lines 34-37** of the introduction).

Comment 2.10: The introduction about the role and abundance of terpenes is rather general and the text part up to the description of the actual experiments should be shortened considerably. On the other hand, the necessity of diversity approaches is discussed without comparatively mentioning existing alternatives.

Response: We have now revised and shortened the introduction to streamline the message of this manuscript, in agreement also with the comment of reviewer #1 (**lines 1-47**).

Regarding mention to existing alternatives, these are now discussed both in the introduction (**lines 34-37**) in response to comments 2.1 and 2.9, and also in the discussion (**lines 490-507**), where biotechnological approaches are discussed. However, some general background about terpenoid biosynthesis is still required because of the broad audience of NCOMMS.

Comment 2.11: The selection of terpene synthases was limited to those forming five- or six-membered ring systems. Chemically, synthases that generate more complex oligocyclic products are of greater interest because, on the one hand, they are often more easily accessible biotechnologically than by chemical means and, on the other hand, they have a broader biological activity profile.

Response: This is a very good suggestion and we are very interested in exploring such terpene synthases in the future.

Comment 2.12: Overall, this is conceptually a very interesting paper. However, the quality in terms of handling and analysis of the products has been unsatisfactorily executed. The authors should devote considerable greater care to the chemical aspects of their work, since the motivation for this research is solely to produce new terpenoids that would not otherwise be readily available, especially with respect to the handling of unnatural diphosphate-containing substrates.

Response: Thank you for highlighting the novelty and significance of the concept! We have now devoted considerable effort in further compound characterization and, in the revised version, we are providing substantially more chemical structures and solid information for the identity of the new building blocks.

Reviewer #3:

Comment 3.1: The manuscript by Ignea et al. entitled “Expanding the terpene biosynthetic code with non-canonical 16 carbon atom building blocks” is a well written description of a study in which the authors engineer yeast to produce non-canonical sesquiterpene derivatives in an effort to expand the availability of natural-product like organic compounds. The study follows nicely work from the same group in which a similar approach was used to generate methylated derivatives of monoterpenes. They begin by engineering yeast to express wild-type and mutant forms of a methyltransferase from the rhizobacterium *Serratia plymuthica*, SpSodMT, that produce novel cyclic, methylated sesquiterpene derivatives as substrates for two clerodane type diterpenoid synthases, terpentetriene synthase (Cyc2) from *Kitasatospora griseola*, and kolavelool synthase (HaKS) from *Herpetosiphon aurantiacus*. Finally, they complete the study by introducing a P-450, abietadiene oxidase (CYP720B1) from *Pinus taeda*, for oxidative adornment of the modified sesquiterpene scaffolds. All told they generate more than 60 new compounds that the authors coin “novoterpenes”. The study is well-designed and executed. The results represent a clear advance in the field. The work is timely and will appeal to the broad readership of Nature Communications.

Response: Thank you for the very positive evaluation of our work. We hope that, with the additional experiments included in the revised version, this manuscript will indeed make a clear advance in the field.

Reviewers' Comments:

Reviewer #2:

Remarks to the Author:

The authors have made significant changes from the first version of the submission. The points raised in my first report have been included, and the authors have done extensive analytical work on the structural elucidation of biotransformation products. They also scaled down their claim that oxidation is an extension of their product library due to the lack of concrete data. This step does decipher the potential of their protocol and concept for STCs and provides guidance to both them and those working in the field for future advancements in the research field. At the very least, examples of oxidation products are now being verified through isolation and analytical work.

I propose to accept the manuscript for publication.

Reviewer #4:

Remarks to the Author:

This manuscript investigates the production of new 16-carbon building blocks via SodMT engineering. SodMT has already been identified in the rhizobacterium *Serratia plymuthica* and heterologously expressed in *E. coli* (Duell et al. 2019. *Microbial cell factories*, 18(1), pp.1-11.). The biosynthesis of C16 prenyl diphosphates through methyltransferases has been previously demonstrated (e.g., Drummond et al. 2019. *ACS Synth. Biol.* 8, 1303-1313). Moreover, a detailed site-directed mutagenesis study of SodMT has been previously performed, in which amino acid residues responsible for the enzymatic activity of SodMT were identified. In addition, some of the synthesized SodMT mutants showed the production of new compounds (Lemfack et al. 2021. *Scientific reports*, 11, 1-13). Considering the aforementioned literature, the current study is not original enough to be published in *Nature Communications*.

In the revised version of the manuscript, the authors have elucidated further compounds to confirm the synthesis of non-canonical isoprenoid building blocks. The structures of the remaining 41 compounds were not elucidated. As mentioned in the first review, these compounds are assumed to possess a C16 carbon framework based on their low-resolution MS spectra. After careful analysis of the chromatograms, many of these compounds correspond to peaks that appear near the baseline. This affects the quality of the MS spectra and may lead to the wrong conclusions. Further structural data are necessary to support the proposed assignments of the remaining 41 compounds.

In Figure 4, chromatogram b, the authors assigned the peak at 13.9 min to 4b. However, the major product 4a is missing. The same applies to Supplementary Figure 5e. The authors assigned the peak at 15.4 min to 4c. This should mean that precursor 4 (WPP) is formed. However, the expected major metabolites 4a and 4b are missing from this chromatogram.

In Supplementary Figure 6, the authors mention that compounds 4a and 4b were detected by selecting the specific ion m/z 109. However, in Supplementary Table 2, the characteristic peak for building block 4 is assigned as m/z 82. Moreover, the characteristic peak for building blocks 5 and 6 is the same (m/z 123). How could the authors differentiate between the minor compounds 6b-6d, 5d, and 5e compounds whose structures have not been elucidated? There is considerable MS signal overlap among the different building blocks, which can lead to peak misassignments.

In the revised version, the authors conducted *in vitro* experiments to support the *in vivo* data. However, it should be noted that the *in vitro* experiments were performed using extracts from yeast cells expressing SpSodMT or selected variants. In yeast lysates, the produced compounds might be subject to derivatization by native enzymes. Moreover, there are several yeast metabolites in the chromatograms that overlap with the peaks of interest. Therefore, it is not surprising that the

obtained chromatograms of the in vitro experiments show the same complexity as those of the in vivo experiments. In addition, many of the assigned building blocks in the in vivo experiments do not match those of the in vitro experiments, which raises the question of whether some of the building blocks are genuine enzymatic products. For example, in Figure 4g, 5 (BPP) was identified as the main building block along with seven minor building blocks. On the other hand, in the corresponding chromatogram from the in vitro experiments (Supplementary Figure 9), the main building block appears to be 4 (WPP) and only four more building blocks were identified. In vitro experiments with purified enzymes would be more appropriate for a clearer view of the enzymatic products and relative correlation.

In the 'Response to Reviewers' comments (response to comment 1.4), the authors argued: 'the promiscuous activity of Cyc2 and HaKS was known, but it was regarding C20 substrates. This was one of the reasons these two enzymes were among those selected for this study. The text was adjusted to make this more obvious and the proposed citation was also added in line 318'.

This statement is not correct. Cyc2 has been shown to catalyze the conversion of the C15 farnesyl diphosphate to various farnesene isomers. This suggests that Cyc2 acts on substrates of various length (e.g., C15 and C20). In fact, the authors mention that in lines 328-330 and provide the respective citations (40 and 44).

There are mistakes in Figure 5A. The chromatograms belonging to SpSodMT(F58L) and SpSodMT(Q57N) should be exchanged.

Line 361: This statement is wrong. In Figure 5a and Supplementary Figure 12, the products are considerably different from the building blocks presented in Figure 4 and Supplementary Figure 8.

Line 382: In Figure 5b, only the products 1b-4b are shown.

Lines 436-438. This is not the case according to Supplementary Figure 15. The intensity of the peak corresponding to 2b is not reduced, suggesting that the conclusion about the production of 33 is wrong.

Figure 6e is not supported by the data. The structures of compounds 30 – 39 have not been characterized and conclusions regarding their synthesis should be avoided. The same applies to Figure 6 legend (lines 444 – 459).

The relative ratio of the peaks in Supplementary Figure 15 is different from that in Figure 6 (for the respective chromatograms), which is concerning regarding the validity of the conclusions in the manuscript.

According to Supplementary Figure 15, the enzyme HaKS is not necessary for the production of 13OH-1b and 9OH-3b. These products are formed using the SpSodMT variants and PtAO only.

In Figure 3a, the stereochemistry of the identified building blocks is shown, but nothing is mentioned about the absolute configuration. Another concerning issue regarding the relative configuration is that most of the 1D and 2D-NOESY spectra are of bad quality.

The olfactory properties section is not adequately discussed.

Response to the comments of reviewer #4 (NCOMMS-20-51374)

We would like to thank reviewer #4 for the constructive comments, which we have taken seriously into consideration and addressed thoroughly with additional experiments when possible. Please find below our point-by-point response.

Reviewer #4 (Remarks to the Author):

Comment 4.1: This manuscript investigates the production of new 16-carbon building blocks via SodMT engineering. SodMT has already been identified in the rhizobacterium *Serratia plymuthica* and heterologously expressed in *E. coli* (Duell et al. 2019. *Microbial cell factories*, 18(1), pp.1-11.). The biosynthesis of C₁₆ prenyl diphosphates through methyltransferases has been previously demonstrated (e.g., Drummond et al. 2019. *ACS Synth. Biol.* 8, 1303-1313). Moreover, a detailed site-directed mutagenesis study of SodMT has been previously performed, in which amino acid residues responsible for the enzymatic activity of SodMT were identified. In addition, some of the synthesized SodMT mutants showed the production of new compounds (Lemfack et al. 2021. *Scientific reports*, 11, 1-13). Considering the aforementioned literature, the current study is not original enough to be published in *Nature Communications*.

Response: We strongly feel that the studies mentioned by the reviewer do not, in any way, limit the impact or novelty of our work.

1. The work by Duell and co-workers reports the cloning of the 4.6 kb candidate sodorifen biosynthetic gene cluster from *S. plymuthica* WS3236 (containing *SpSodMT* and three more enzymes) and its heterologous expression in *E. coli* to achieve sodorifen production. Our work does not focus on sodorifen production. Rather, it utilizes *SpSodMT* as the basis for an extensive mutagenesis effort to derive nine novel C₁₆ prenyl diphosphate building blocks that are subsequently utilized to synthesize unprecedented terpene scaffolds and further decorated terpenoids with 16 carbons. None of these findings is related to the study of Duell *et al.*

The article by Duell *et al.* and its relevance to our work was already clearly discussed in the “Introduction” section of the original manuscript (**line 61** of current revised version).

2. The work of Drummond and co-workers utilizes an IPP-specific methyltransferase to synthesize several carotenoid compounds that appear to have additional carbon atoms in their skeleton. However, none of these new carotenoids was isolated or structurally

characterized. In the same study, evidence is also presented for the synthesis of 8-methyl-FPP, which, however, was never utilized by a terpene synthase or prenyltransferase to make specific products. In our work, we have isolated and elucidated the structures of 28 new terpenoids obtained by the synthesis of, not only one, but nine different novel methylated forms of FPP. The limitations of the study of Drummond *et al.* were already clearly explained in the “Discussion” section of the original manuscript (**lines 339-354** of current revised version).

3. The study by Lemfack *et al.* aimed to understand the reaction mechanism of *SpSodMT*. In this work, the authors substituted several residues lining the active site cavity with alanine. Most of these alanine substitutions resulted in inactive enzymes but the authors did not conclusively determine whether the resulting activity loss was due to the impairment of protein structure/stability or the direct involvement of the specific residue in the catalytic mechanism. In addition to their main findings, they observed that in the case of two mutants, L239A and C241A, minor new peaks could be detected in gas chromatography / low-resolution MS analysis of the products of *in vitro* reactions. However, none of the compounds corresponding to these minor peaks were isolated or structurally characterized and their relation to PSPP remains unclear. In contrast to the report of Lemfack *et al.*, our work presents a thorough and highly successful site-directed mutagenesis effort that results in the development of efficient *SpSodMT* variants and the production of a large number of isolated in pure form and structurally characterized C₁₆ terpenoids.

In conclusion, the papers mentioned by reviewer #4 are only marginally related to our work and do not take away from the novelty of our findings. In our work, we develop unique *SpSodMT* mutants that catalyze the production of nine new prenyl diphosphate building blocks with 16 carbons, which we subsequently employ as substrates for enzymes catalyzing the downstream steps of terpene biosynthesis. Thus, we achieve to reconstruct a complete orthogonal C₁₆ terpene biosynthetic pathway in yeast cells. This pathway includes all steps required to convert sugar to cytochrome P450-decorated terpenoids. We solidly support our findings with the isolation and structural characterization of 28 C₁₆ terpenoid compounds. Our work clearly takes the field forward in an unprecedented direction by establishing the biosynthesis of a whole new group of non-canonical terpenoid compounds.

Comment 4.2: In the revised version of the manuscript, the authors have elucidated further compounds to confirm the synthesis of non-canonical isoprenoid building blocks. The structures of

the remaining 41 compounds were not elucidated. As mentioned in the first review, these compounds are assumed to possess a C₁₆ carbon framework based on their low-resolution MS spectra. After careful analysis of the chromatograms, many of these compounds correspond to peaks that appear near the baseline. This affects the quality of the MS spectra and may lead to the wrong conclusions. Further structural data are necessary to support the proposed assignments of the remaining 41 compounds.

Response: In response to this comment:

1. We extended our structural analysis effort to identify additional non-canonical C₁₆ compounds. In the revised version, we report a total of 28 structurally characterized molecules derived from the C₁₆ prenyl diphosphates (summarized in revised **Fig. 1**).
2. We removed the tentative assignments of the remaining identified compounds and revised the manuscript text and figures to focus almost exclusively on the 28 compounds we have isolated and elucidated. From the remaining products, we have only retained 3d, 4d, and 11-15 for which we report here high resolution MS data (**new Supplementary Figures 11 and 14**) to confirm their C₁₆ nature.

With the additional data provided and the corresponding changes in the manuscript, we strongly feel that the experimental results reported clearly demonstrate the biosynthesis of non-canonical C₁₆ terpenoids and solidly support the main claims of the manuscript.

Comment 4.3: In Figure 4, chromatogram b, the authors assigned the peak at 13.9 min to 4b. However, the major product 4a is missing. The same applies to Supplementary Figure 5e. The authors assigned the peak at 15.4 min to 4c. This should mean that precursor 4 (WPP) is formed. However, the expected major metabolites 4a and 4b are missing from this chromatogram.

Response: In Fig. 4, compound 4a was initially not shown because it was minor (when yeast strains are grown in non-buffered media, the main product is the tertiary alcohol). In response to this comment, we now provide a revised Fig. 4 that includes annotation of the minor peak corresponding to 4a.

In Supplementary Figure 5, the assignment of the 15.4 min peak as 4c was a mistake in the annotation of the chromatogram. This has been corrected. Thank you for pointing this out.

Comment 4.4: In Supplementary Figure 6, the authors mention that compounds 4a and 4b were detected by selecting the specific ion m/z 109. However, in Supplementary Table 2, the

characteristic peak for building block 4 is assigned as m/z 82. Moreover, the characteristic peak for building blocks 5 and 6 is the same (m/z 123). How could the authors differentiate between the minor compounds 6b-6d, 5d, and 5e compounds whose structures have not been elucidated? There is considerable MS signal overlap among the different building blocks, which can lead to peak misassignments.

Response: In response to this comment and comment 4.2 above, we have revised the manuscript text and figures to remove assignments of compounds whose structures have not been elucidated. Therefore, tentative assignments of all the compounds mentioned in this comment have been removed.

Comment 4.5: In the revised version, the authors conducted *in vitro* experiments to support the *in vivo* data. However, it should be noted that the *in vitro* experiments were performed using extracts from yeast cells expressing SpSodMT or selected variants. In yeast lysates, the produced compounds might be subject to derivatization by native enzymes. Moreover, there are several yeast metabolites in the chromatograms that overlap with the peaks of interest. Therefore, it is not surprising that the obtained chromatograms of the *in vitro* experiments show the same complexity as those of the *in vivo* experiments. In addition, many of the assigned building blocks in the *in vivo* experiments do not match those of the *in vitro* experiments, which raises the question of whether some of the building blocks are genuine enzymatic products. For example, in Figure 4g, 5 (BPP) was identified as the main building block along with seven minor building blocks. On the other hand, in the corresponding chromatogram from the *in vitro* experiments (Supplementary Figure 9), the main building block appears to be 4 (WPP) and only four more building blocks were identified. *In vitro* experiments with purified enzymes would be more appropriate for a clearer view of the enzymatic products and relative correlation.

Response: We have tried to perform these experiments with enzymes expressed and purified from *E. coli*. However, although it was possible to obtain some activity from the wild-type enzyme (Supplementary Fig. 4), the stability/activity of the *E. coli*-produced mutants was not sufficient for *in vitro* activity assays. For this reason, we resorted in using yeast cell extracts for these assays. Despite the objection of reviewer #4, the results obtained from the yeast-produced enzymes do support the main claims made in the manuscript in terms of the product selectivity of the mutants, which was their intended purpose (these experiments were carried out in response to a comment by reviewer #1).

Regarding the low levels of products of building block 5 (BPP), this is likely due to the instability of 5b in the strong acidic conditions used to dephosphorylate the products for analysis by GC-MS. We noticed that in all *SpSodMT* mutants studied using *in vitro* assays, the levels of 5b detected were very low and consistently lower than what was observed in the corresponding *in vivo* experiments. We also observed poor stability of the isolated compound 5b during storage. For this reason, we avoided to conclude in the manuscript that variant *SpSodMT*(F58M-L302S) is selective for BPP.

Overall, we feel that, with the exception of the c-type compounds, it is unlikely that any of the other 28 compounds isolated and structurally characterized is produced by modification of another product by yeast enzymes. As explained in the manuscript and in Supplementary Figure 1, c-type compounds are biosynthesized by reduction of the corresponding primary alcohols by enzymes such as *Oye2p*. However, the remaining compounds are primary and tertiary alcohols that conform well to the proposed enzymatic mechanism in Fig. 3a.

Comment 4.6: In the ‘Response to Reviewers’ comments (response to comment 1.4), the authors argued: ‘the promiscuous activity of *Cyc2* and *HaKS* was known, but it was regarding C20 substrates. This was one of the reasons these two enzymes were among those selected for this study. The text was adjusted to make this more obvious and the proposed citation was also added in line 318’. This statement is not correct. *Cyc2* has been shown to catalyze the conversion of the C15 farnesyl diphosphate to various farnesene isomers. This suggests that *Cyc2* acts on substrates of various length (e.g., C15 and C20). In fact, the authors mention that in lines 328-330 and provide the respective citations (40 and 44).

Response: To describe the activity of these enzymes more accurately, the text in **line 242** of the manuscript was edited to “substrates of various length”.

Comment 4.7: There are mistakes in Figure 5A. The chromatograms belonging to *SpSodMT*(F58L) and *SpSodMT*(Q57N) should be exchanged.

Response: Thank you for pointing this out. The labeling of the chromatograms was inadvertently switched during the previous revision. This has now been corrected.

Comment 4.8: Line 361: This statement is wrong. In Figure 5a and Supplementary Figure 12, the products are considerably different from the building blocks presented in Figure 4 and Supplementary Figure 8.

Response: We believe this comment is probably a misunderstanding. The products are different between Fig. 4 and Fig. 5 because in the latter figure, Cyc2 is co-expressed promoting production of 1d, 2d, 3d and 4d from the corresponding substrates.

Comment 4.9: Line 382: In Figure 5b, only the products 1b-4b are shown.

Response: Correctly pointed out. This has now been rephrased (**lines 278-279** of the current revised manuscript).

Comment 4.10: Lines 436-438. This is not the case according to Supplementary Figure 15. The intensity of the peak corresponding to 2b is not reduced, suggesting that the conclusion about the production of 33 is wrong.

Response: In the revised version, we have now removed most claims relating to compounds whose structure has not been elucidated and, as a result, this comment has also been removed (revised text in **lines 307-308**).

Comment 4.11: Figure 6e is not supported by the data. The structures of compounds 30 – 39 have not been characterized and conclusions regarding their synthesis should be avoided. The same applies to Figure 6 legend (lines 444 – 459).

Response: As already discussed in response to previous comments, we have now edited the text to remove any conclusions related to the origin of compounds not isolated in pure form and fully characterized (new text in **lines 300-312**).

Comment 4.12: The relative ratio of the peaks in Supplementary Figure 15 is different from that in Figure 6 (for the respective chromatograms), which is concerning regarding the validity of the conclusions in the manuscript.

Response: We realized that the fact that we previously chose to show the experiment carried out in non-buffered media made interpretation of Figures 6 and S15 confusing. Therefore,

we repeated the experiments shown in Figure 6, including the controls, which were shown previously in Supplementary Figure 15, this time using buffered media to minimize production of b-type compounds in cells lacking HaKS. We now present a **revised Fig. 6** that includes all the control experiments together clearly showing the specific production of PtAO-derived compounds.

Comment 4.13: According to Supplementary Figure 15, the enzyme HaKS is not necessary for the production of 13OH-1b and 9OH-3b. These products are formed using the SpSodMT variants and PtAO only.

Response: As already discussed in the manuscript and in the response to reviewer #1 (comment 1.4), the tertiary alcohols (b-type compounds) are also produced by acid hydrolysis in non-buffered yeast media. This complicated the interpretation of these results. As discussed above, we have replaced Figures 6 and S15 with a **new Fig. 6** that presents the results obtained using buffered media to minimize the background reaction.

Comment 4.14: In Figure 3a, the stereochemistry of the identified building blocks is shown, but nothing is mentioned about the absolute configuration. Another concerning issue regarding the relative configuration is that most of the 1D and 2D-NOESY spectra are of bad quality.

Response: Indeed, only the relative configuration of the isolated compounds has been determined. The absolute stereochemistry through single crystal X-ray diffraction analysis was not possible to determine due to the oily nature of the isolated metabolites. In addition, these metabolites lacked significant chromophores to allow the determination of absolute configuration through CD analysis. Furthermore, the lack of any tertiary alcohol (with the exception of 9OH-3b) did not allow the use of derivatization with Mosher's reagent. However, this does not affect the significance of our findings, which does not really pertain to the absolute configuration of the produced and characterized C16 compounds.

Regarding the quality of the 1D and 2D-NOESY spectra, taking into account the amounts in which the new C16 compounds were isolated in and the field that was available, it is sufficient in our opinion to determine with certainty and beyond any doubt the relative configuration in all stereogenic centers of the isolated compounds.

Comment 4.15: The olfactory properties section is not adequately discussed.

Response: We have now extended the analysis of the olfactory properties on the new compounds by focusing exclusively on the 28 isolated and characterized compounds. We have revised the corresponding section to present the new findings (**lines 317-324**) and replaced previous Figure 7 with a table (**Table 1**) describing the main odor descriptors of the four identified odor-active molecules.

Reviewers' Comments:

Reviewer #4:

Remarks to the Author:

The manuscript describes the production of presodorifenyl diphosphate analogs through protein engineering of a methyltransferase (SpSodMT). SpSodMT is a bifunctional enzyme that combines the function of a methyltransferase with that of a terpene synthase. It should be noted that there is no need for sequential addition of HaKS/Cyc2 and PtAO enzymes to produce new building blocks and oxygenated compounds (as shown in Figure 1). The same products can also be identified in the chromatograms of SpSodMt wild type or variants by acid hydrolysis or endogenous yeast enzymes. Regarding the response to comment 4.2: The authors have revised the manuscript and figures to remove assignments of compounds whose structures have not been elucidated. However, there are still issues that need to be addressed. In Figure 4f, the authors assigned a peak as a scaffold 6-derived compound. How do the authors know that this compound belongs to building block 6? It could be an isomer with a similar mass spectrum. Was it also used for quantification? This is an important issue as the authors concluded that the respective variant SpSodMT (F58M-L302Q) produces predominantly the building block 6. Similarly, in Figure 6, the authors suggest that the uncharacterized compounds 11-15 are building block 2-derived products. If this is correct, why compounds 14 and 15 are not detected in the chromatogram of SpSodMT(F58L) (Figure 6c), which also produces building block 2? Compounds 11-15 could be derived from other building blocks produced in the respective SpSodMT variants.

Regarding the response to comment 4.3: In the revised Figure 4b, the retention time of 4b is not the same as that in the chromatograms of Figures 4d-g. The authors should check the correct assignments of the peaks.

Regarding the response to comment 4.5: The authors mentioned that they avoided to conclude in the manuscript that variant SpSodMT(F58M-L302S) is selective for BPP. However, in the text (page 12, line 203), the authors suggested that SpSodMT(F58M-L302S) produces BPP preferentially.

Regarding the response to comment 4.13: In the revised Figure 6, the authors used buffered media to minimize the production of b-type compounds in cells lacking HaKS. However, in non-buffered yeast media, the enzyme HaKS is not necessary for the production of 13OH-1b or 9OH-3b. These products can also be formed using the SpSodMT variants and PtAO only (previously shown in the Supplementary Figure 15, deleted in the revised version). The same applies to Cyc2 and HaKS enzymatic products (1d-4d and 1b-4b, respectively). These products can also be identified in the chromatograms of SpSodMt wild type or variants, as mentioned above.

Regarding the response to comment 4.14: The stereochemistry of the identified building blocks is important as some are stereoisomers (e.g., building blocks 5 and 8). The authors did not justify how they assigned the absolute stereochemistry of the various building blocks without performing any additional experiments.

In Supplementary Table 9, the titers of 2b and 4b compounds are provided for the strain co-expressing HaKS and SpSodMt wild-type. This suggests that the SpSodMt wild-type enzyme produces the new building blocks 2 and 4, which is not possible according to the authors' conclusions. Such mistakes raise questions about the validity of the data.

In Supplementary Table 11, the titer of the minor compound 13OH-1b (69.62 mg/L) is higher than that of the major product 1b (59.3 mg/L, Supplementary Table 9). This should not be the case considering the ratio of the 1b and 13OH-1b peaks (Figure 6a chromatogram). The same applies to 9OH-3b. The authors should correct the titer values in the corresponding tables and text.

NCOMMS-20-51374B - response to reviewer #4:

Comment 1: The manuscript describes the production of presodorifenyl diphosphate analogs through protein engineering of a methyltransferase (SpSodMT). SpSodMT is a bifunctional enzyme that combines the function of a methyltransferase with that of a terpene synthase. It should be noted that there is no need for sequential addition of HaKS/Cyc2 and PtAO enzymes to produce new building blocks and oxygenated compounds (as shown in Figure 1). The same products can also be identified in the chromatograms of SpSodMt wild type or variants by acid hydrolysis or endogenous yeast enzymes.

Response: The main aim of the manuscript is to recapitulate the modular structure of terpene biosynthesis on C₁₆ precursors by establishing the synthesis of non-canonical prenyl diphosphate building blocks that can then be further converted into specific novel C₁₆ terpenoids. We start by constructing SpSodMT mutants that synthesize nine new prenyl diphosphate building blocks. To convert these building blocks into terpene scaffolds, we utilize a combination of methods. Synthesis of the corresponding primary alcohols is done by taking advantage of the well-documented activity of endogenous yeast phosphatases (Dpp1p and Lpp1p) on prenyl diphosphates^{1,2}. Formation of the 2,3-dihydro form of the primary alcohols utilizes the yeast endogenous Oye2p oxidoreductase³. Synthesis of the corresponding tertiary alcohols is achieved in this work by two methods. One method used involves using the well-documented observation that when excess prenyl diphosphates are produced in yeast cells, these are released to the culture medium where they become hydrolyzed because of the acidic environment⁴. This method is less efficient and leads to a mixture of products (also including different olefins). The second method used here was to identify a terpene synthase (HaKS) that take these new prenyl diphosphates as substrates and produce the corresponding tertiary alcohols. This method is far more efficient and results in much higher product titers and a narrower product profile. Formation of the corresponding olefins is done by identifying Cyc2 as a suitable synthase. The efficiency of HaKS and Cyc2 catalyzed production is about 10 times higher than acid-hydrolysis-based production. Indicatively, with wild-type SpSodMT, 1d is produced at 10.3 mg/L when Cyc2 is present but at 1.1 mg/L by acid hydrolysis; 2b is produced by SpSodMT(Q57N) at 33.7 mg/L when HaKS is present and 2.9 mg/L by acid hydrolysis). Therefore, albeit not essential, the use of an enzymatic step is highly advantageous. First, it facilitates the isolation and structural analysis of the obtained products. Second, it enables the efficient bioproduction of the target compounds, which may have industrial applications. Furthermore, identifying enzymes able to convert the new substrates expands the synthetic biology toolbox and makes the system less dependent on the host and more predictable compared to acid hydrolysis. Further screening for suitable terpene synthases or protein engineering of identified enzymes will lead to additional unique structures and further expand the system.

Comment 2: Regarding the response to comment 4.2: The authors have revised the manuscript and figures to remove assignments of compounds whose structures have not been elucidated. However, there are still issues that need to be addressed. In Figure 4f, the authors assigned a peak as a scaffold 6-derived compound. How do the authors know that this compound belongs to building block 6? It could be an isomer with a similar mass spectrum. Was it also used for quantification? This is an important issue as the authors concluded that the respective variant SpSodMT (F58M-L302Q) produces predominantly the building block 6. Similarly, in Figure 6, the authors suggest that the uncharacterized compounds 11-15 are building block 2-derived products. If this is correct, why compounds 14 and 15 are not detected in the chromatogram of SpSodMT(F58L) (Figure 6c), which also produces building block 2? Compounds 11-15 could be derived from other building blocks produced in the respective SpSodMT variants.

Response: In the revised version, we decided to remove all unassigned peaks except a 6-derived compound (likely, but not confirmed, 6b), 3d and 4d, and the PtAO-derived products 11-15. This was done because we have sufficient data to confirm their C₁₆ nature, despite not having the NMR data. The assignment of the peak in Fig 4f as a 6-derived compound was made based on the fact that when compared across the panel of different mutants analyzed, this peak is present only when 6a (an isolated and elucidated compound) is present, and absent when each of the remaining building blocks is produced but not 6. This can be done by comparing the profiles of SpSodMT(F58M), SpSodMT(V273A), SpSodMT(L302Q), SpSodMT(L302S), SpSodMT(F58M-V273A), and SpSodMT(F58M-L302Q). The EI mass-spectrum and relative GC retention time are also consistent with this peak being 6-derived (Supplementary Table 3). This 6-derived compound was indeed included in the quantification but its inclusion does not alter the main conclusion. Building block 6 is the predominant product regardless of this peak being included or not.

In Fig 6, compounds 14 and 15 are not indicated in panel c for simplicity because they are in minor amounts. We can confirm that these compounds are indeed synthesized in this experiment using selected fragment ions. We think that it is reasonable that compounds 14 and 15 are synthesized in lower amounts in the experiment in panel c (compared to the one in panel b) because the combination of HaKS with SpSodMT(F58L) is four-times less efficient than HaKS and SpSodMT(Q57N) in the production of 2b/2a (Supplementary Table 9).

Comment 3: Regarding the response to comment 4.3: In the revised Figure 4b, the retention time of 4b is not the same as that in the chromatograms of Figures 4d-g. The authors should check the correct assignments of the peaks.

Response: Thank you pointing this out. Figure 4b has now been corrected.

Comment 4: Regarding the response to comment 4.5: The authors mentioned that they avoided to conclude in the manuscript that variant SpSodMT(F58M-L302S) is selective for BPP. However, in the text (page 12, line 203), the authors suggested that SpSodMT(F58M-L302S) produces BPP preferentially.

Response: We think this may be a misunderstanding. The sentence in page 12 that the reviewer refers to discusses yeast-based production. The conclusion is correct in this context because the corresponding yeast strain produces preferentially BPP-derived products. Our response to comment 4.5 of the previous revision referred to the *in vitro* experiment. In the section in lines **203-208**, where the *in vitro* experiments are discussed, we indeed avoided to conclude this selectivity.

To avoid any further confusion, we have removed discussion of the yeast result from **line 202**.

Comment 5: Regarding the response to comment 4.13: In the revised Figure 6, the authors used buffered media to minimize the production of b-type compounds in cells lacking HaKS. However, in non-buffered yeast media, the enzyme HaKS is not necessary for the production of 13OH-1b or 9OH-3b. These products can also be formed using the SpSodMT variants and PtAO only (previously shown in the Supplementary Figure 15, deleted in the revised version). The same applies to Cyc2 and HaKS enzymatic products (1d-4d and 1b-4b, respectively). These products can also be identified in the chromatograms of SpSodMt wild type or variants, as mentioned above.

Response: This has been answered above (comment 1).

Comment 6: Regarding the response to comment 4.14: The stereochemistry of the identified building blocks is important as some are stereoisomers (e.g., building blocks 5 and 8). The authors did not justify how they assigned the absolute stereochemistry of the various building blocks without performing any additional experiments.

Response: As also mentioned in our previous communication with the reviewer, the absolute stereochemistry of the isolated compounds has not been determined and thus no relevant comment has been included in the text nor in the supplementary note pertaining to structure elucidation. Figure S1 depicts the relative stereochemistry of the chiral centers of the isolated metabolites that has been determined on the basis of the NOE enhancements observed in 1D NOE and/or 2D NOESY experiments (please see the relevant Figure S3). For example, the relative stereochemistry of building block **5** has been assigned as 6*S**,8*S**,11*S**, while the relative stereochemistry of building block **8** has been determined as 6*S**,8*S**,11*R**. Similarly, the relative stereochemistry of both building blocks **4** and **7** were determined as 6*R**,7*R**,11*S**.

Although it would have been a nice addition to be able to determine the absolute stereochemistry of the isolated metabolites, single crystal X-ray diffraction analysis was not possible due to oily nature of the

isolated metabolites. Furthermore, CD analysis was not attempted since the isolated compounds lacked any significant chromophores, whereas the lack of any tertiary alcohol did not allow derivatization with Mosher's reagent. Nonetheless, we strongly believe that the fact that the absolute stereochemistry of the isolated compounds was not determined does not diminish the significance of our findings, which focus on the establishment of the biosynthesis of a new group of non-canonical terpenoid compounds.

Comment 7: In Supplementary Table 9, the titers of 2b and 4b compounds are provided for the strain co-expressing HaKS and SpSodMt wild-type. This suggests that the SpSodMt wild-type enzyme produces the new building blocks 2 and 4, which is not possible according to the authors' conclusions. Such mistakes raise questions about the validity of the data.

Response: This is not a mistake. On the contrary, as explained below, this observation shows that the system we have developed is efficient and only adds to the validity of the data presented.

The reason that these two products are visible is that SpSodMT does not only make PSPP but also minor amounts (less than 1%) of 2 and 4. This ability of SpSodMT to make such low amounts of other products was not possible to observe in previous publications reporting the characterization of SpSodMT, likely because the system used was less sensitive/efficient (see for example Fig. 2 in von Reuss et al. ⁵). However, here we are able to observe these products because we are using HaKS, which is able to utilize the minor levels of 2 and 4 and synthesize sufficient amounts of 2b and 4b to be detected (when acid hydrolysis was used, these products were not readily visible because these reactions were inefficient; see response to comment 1). This low-level promiscuity of SpSodMT is not unexpected considering the similarity of the later steps of its reaction mechanism with the mechanism of terpene synthases.

We avoided to discuss this observation in the manuscript because of size and cohesion considerations and because it does not add to the main findings of this work. But we have added a note in **Supplementary Table 9** to explain why these compounds are present.

Comment 8: In Supplementary Table 11, the titer of the minor compound 13OH-1b (69.62 mg/L) is higher than that of the major product 1b (59.3 mg/L, Supplementary Table 9). This should not be the case considering the ratio of the 1b and 13OH-1b peaks (Figure 6a chromatogram). The same applies to 9OH-3b. The authors should correct the titer values in the corresponding tables and text.

Response: Thank you for pointing this out. We have repeated the quantification and updated **Supplementary Table 11** accordingly.

References

- 1 Ignea, C., Pontini, M., Maffei, M. E., Makris, A. M. & Kampranis, S. C. Engineering monoterpene production in yeast using a synthetic dominant negative geranyl diphosphate synthase. *ACS Synth Biol* **3**, 298-306, doi:10.1021/sb400115e (2014).
- 2 Faulkner, A. *et al.* The LPP1 and DPP1 gene products account for most of the isoprenoid phosphate phosphatase activities in *Saccharomyces cerevisiae*. *J Biol Chem* **274**, 14831-14837 (1999).
- 3 Brown, S., Clastre, M., Courdavault, V. & O'Connor, S. E. De novo production of the plant-derived alkaloid strictosidine in yeast. *Proc Natl Acad Sci U S A* **112**, 3205-3210, doi:10.1073/pnas.1423555112 (2015).
- 4 Goodman, D. S. & Popjak, G. Studies on the biosynthesis of cholesterol. XII. Synthesis of allyl pyrophosphates from mevalonate and their conversion into squalene with liver enzymes. *J Lipid Res* **1**, 286-300 (1960).
- 5 von Reuss, S. *et al.* Sodorifen Biosynthesis in the Rhizobacterium *Serratia plymuthica* Involves Methylation and Cyclization of MEP-Derived Farnesyl Pyrophosphate by a SAM-Dependent C-Methyltransferase. *J Am Chem Soc* **140**, 11855-11862, doi:10.1021/jacs.8b08510 (2018).

Reviewers' Comments:

Reviewer #4:

Remarks to the Author:

Comment to Response 1: As the authors mentioned in their response, the use of terpene synthases and cytochrome P450 oxidases in this paper was advantageous, albeit not essential.

Comment to Response 2: The authors responded that the assignment of the peak in Fig 4f as a 6-derived compound was made based on the fact that when compared across the panel of different mutants analyzed, this peak is present only when 6a is present. However, this is not correct. In the chromatograms of SpSodMT(V273A) and SpSodMT(F58L), peak 6b is missing even though 6a is present. In these chromatograms, peak 6a is present in considerable amounts, and therefore peak 6b should be present in relatively similar intensity, as in the case of Fig 4f and SpSodMT(L302S) (Supplementary Figure 5e).

Regarding compounds 14 and 15: In Fig. 6b, compound 14 has the highest intensity among compounds 11-15. The same trend should be expected for Fig. 6c. However, only the minor compounds 11-13 are present in Fig. 6c. This discrepancy in the chromatograms suggests that the uncharacterized compounds 11-15 are not building block 2-derived products. The origin of compounds 11-15 needs further investigation.

Comment to Response 3: The retention time of peak 4b has been corrected in Figure 4b. However, the magnified region showing this peak does not correspond to the chromatogram of SpSodMT(Q57N). The authors should provide the correct region.

Comment to Response 4: The authors should further discuss that they could not confirm that the variant SpSodMT(F58M-L302S) is selective for BPP using in vitro assays, as they did for SpSodMT(F58V) (page 12, lines 252-254), and propose possible explanations.

Comment to Response 6: It is clear that the authors cannot determine the absolute configuration. However, the authors present structures with absolute configurations in the paper. Please add a comment that the absolute configuration of the compounds was tentatively assigned.

Comment to Response 7: The authors mention that the ability of SpSodMT to make such low amounts of other products was not possible to observe in previous publications reporting the characterization of SpSodMT, likely because the system used was less sensitive/efficient. This is misleading, as the authors also did not identify any other products (i.e., 2 and 4) in the chromatogram of wild-type SpSodMT (Figure 4a and Supplementary Table 5). The only time the authors identified 2 and 4 was in the chromatogram of the strain co-expressing HaKS and SpSodMt wild-type. However, in the corresponding chromatogram (Figure 5b), not even traces of these compounds can be seen. The amount of 2b (0.60 mg/L) is high enough for the corresponding peak to appear in the chromatogram. The fact that the wild-type SpSodMT produces scaffolds 2 and 4 is important and contradicts the authors' conclusion that they are new to nature. The authors should further investigate and discuss the production of those building blocks from the wild-type SpSodMT (in the text and not as a note in the Supplementary Information). The quantities are low but in the same range as scaffolds 7-9, which could be easily identified and characterized.

Comment to Response 8: The authors should update the corrected values in the text as well.

Response to reviewer (NCOMMS-20-51374C)

We would like to thank reviewer #4 for the useful comments. Please find below our point-by-point response.

Comment to Response 1: As the authors mentioned in their response, the use of terpene synthases and cytochrome P450 oxidases in this paper was advantageous, albeit not essential.

Response: Although we understand that with this response the reviewer considers this issue concluded, we feel obliged to clarify again that the use of cytochrome P450 oxidases was essential and not advantageous.

Comment to Response 2 (Part 1): The authors responded that the assignment of the peak in Fig 4f as a 6-derived compound was made based on the fact that when compared across the panel of different mutants analyzed, this peak is present only when 6a is present. However, this is not correct. In the chromatograms of SpSodMT(V273A) and SpSodMT(F58L), peak 6b is missing even though 6a is present. In these chromatograms, peak 6a is present in considerable amounts, and therefore peak 6b should be present in relatively similar intensity, as in the case of Fig 4f and SpSodMT(L302S) (Supplementary Figure 5e).

Response: First, we would like to clarify that this 6-derived compound is also present in the chromatograms of SpSodMT(V273A) and SpSodMT(F58L) but in much lower levels. These chromatograms were not labelled for simplicity.

We consider that the assignment as a 6-derived compound is correct because when 6a is absent, this compound is also absent. This 6-derived compound was not stable and for this reason, we were not able to isolate and characterize it. Because of this instability, its levels vary between samples. This is not surprising, as we are using a biological system and there is variation originating from the culture conditions, extraction steps, etc. Therefore, in our opinion, using the relative levels of this compound as evidence to infer (or not) its origin, as done by the reviewer, cannot be conclusive.

As discussed in our previous response, the identity of this peak is not central to the findings of the manuscript and does not change its main conclusions. To help resolve this discussion, we have removed the contribution of this peak from the product profile of the *SpSodMT* mutants in **Figure 4, Suppl. Figure 5** and **Suppl. Figure 8** and added a comment in the corresponding figure legends explaining that: “The peak indicated with “**” corresponds to an uncharacterized compound possibly derived from 6, as suggested by the observation that it is present only in samples where 6a is also present. The area of this peak was not included in the calculation of the product profile of the *SpSodMT* mutants”. We have also avoided any final conclusions regarding the product preference of *SpSodMT*(F58M-L302Q) or *SpSodMT*(F58M-L302S) in the section in lines 197-207 because of the instability of some of the products.

Comment to Response 2 (Part 2): Regarding compounds 14 and 15: In Fig. 6b, compound 14 has the highest intensity among compounds 11-15. The same trend should be expected for Fig. 6c. However, only the minor compounds 11-13 are present in Fig. 6c. This discrepancy in the chromatograms suggests that the uncharacterized compounds 11-15 are not building block 2-derived products. The origin of compounds 11-15 needs further investigation.

Response: In the previous version of the manuscript, we explained in the text (lines 304-306) that the assignment of compounds 11-15 was tentative because of the lack of structural information. The suggestion that these compounds were derived from building block 2 (shown in the previous versions of Figures 1 and 6d) was based on the fact that they appear only when 2 is present. As described in the response above, because of the biological nature of the system, relative abundance of compounds is not in this case a reliable measure of origin. Therefore, in our opinion, variation in the relative abundance of compound 14 cannot be used to infer the origin of compounds 11-15.

Nevertheless, the main conclusion of this section of the manuscript is that PtAO can decorate C₁₆ terpenoids. This is clearly demonstrated by the isolation and characterization of 13OH-1b and 9OH-3b. The identity of the building block giving rise to 11-15 is not essential for this conclusion. Therefore, in order to avoid any possible doubt and help conclude this discussion, we removed any comments

connecting any of compounds 11-15 with a specific building block. To this end, we modified **Figures 1** and **Figure 6** (and the corresponding figure legends) by removing mention to the origin of compounds 11-15, and rephrased the manuscript text in **lines 304-307** for clarity.

Comment to Response 3: The retention time of peak 4b has been corrected in Figure 4b. However, the magnified region showing this peak does not correspond to the chromatogram of SpSodMT(Q57N). The authors should provide the correct region.

Response: This is probably misunderstood by the reviewer. The inset shows the same region, but selecting for m/z 135, to facilitate the reader. This is indicated in the figure but may have escaped the attention of the reviewer. To make this even clearer, we have now also included an explanation in the corresponding figure legend (**line 954**).

Comment to Response 4: The authors should further discuss that they could not confirm that the variant SpSodMT(F58M-L302S) is selective for BPP using in vitro assays, as they did for SpSodMT(F58V) (page 12, lines 252-254), and propose possible explanations.

Response: In connection to our response to the previous comment (comment to response 2), we have now removed mention to the product preference of SpSodMT(F58M-L302S) from the manuscript and added an explanation for this in lines **205-207**.

Comment to Response 6: It is clear that the authors cannot determine the absolute configuration. However, the authors present structures with absolute configurations in the paper. Please add a comment that the absolute configuration of the compounds was tentatively assigned.

Response: A relevant comment has been added in the legend to Figure 1, as well as in Figure S1 of the Supplementary Note.

Comment to Response 7: The authors mention that the ability of SpSodMT to make such low amounts of other products was not possible to observe in previous publications reporting the characterization of SpSodMT, likely because the system used was less sensitive/efficient. This is misleading, as the authors also did not identify any other products (i.e., 2 and 4) in the chromatogram of wild-type SpSodMT (Figure 4a and Supplementary Table 5). The only time the authors identified 2 and 4 was in the chromatogram of the strain co-expressing HaKS and SpSodMt wild-type.

However, in the corresponding chromatogram (Figure 5b), not even traces of these compounds can be seen. The amount of 2b (0.60 mg/L) is high enough for the corresponding peak to appear in the chromatogram. The fact that the wild-type SpSodMT produces scaffolds 2 and 4 is important and contradicts the authors' conclusion that they are new to nature. The authors should further investigate and discuss the production of those building blocks from the wild-type SpSodMT (in the text and not as a note in the Supplementary Information). The quantities are low but in the same range as scaffolds 7-9, which could be easily identified and characterized.

Response: As a matter of fact, low amounts of 2b and 4b can also be observed in Figure 5b. These were not indicated before for simplicity because we did not consider this matter to be critical for the main conclusions of the manuscript. We have now revised **Figure 5b**, by zooming on these two regions of the chromatogram and selecting m/z to make the presence of 2b and 4b obvious to the reader.

We have also moved the discussion about these two possible minor products in the main manuscript file (**lines 278-282**) instead of the supplementary information, as suggested by the reviewer.

As we explained in the previous response, based on the mechanism of this enzyme, it is not unreasonable that wild-type *SpSodMT* makes minor amounts of other products in the range around or below 1%. This could be a characteristic only of the yeast-produced enzyme or it may also be occurring in *S. plymuthica*. We do not think that this matter is central to the conclusions of this manuscript and we strongly feel that any further investigation of these minor products is well beyond the scope of this manuscript.

Furthermore, regarding the comment of the reviewer about these compounds being new-to-nature, we feel that this had already been clearly answered in the discussion of the previous version (now in lines 359-360 and 374-376 of the revised version). As we explained, although these C₁₆ compounds are identified for the first time here using a synthetic biology approach, it is unclear whether some of them are produced by living organisms but not yet identified (or were produced at some point during evolution). The findings of this manuscript will prompt further investigation of the biosynthesis, evolution, and function of this C₁₆ class.

Comment to Response 8: The authors should update the corrected values in the text as well.

Response: Corrected values have been updated in **lines 302 and 309**.

Reviewers' Comments:

Reviewer #4:

Remarks to the Author:

The authors have addressed the points raised in my review and the revised version is suitable for publication.